# Silica-Based Materials Containing Inorganic Red/NIR Emitters and Their Application in Biomedicine

**DOI:** 10.3390/ma16175869

**Published:** 2023-08-27

**Authors:** Yuri A. Vorotnikov, Natalya A. Vorotnikova, Michael A. Shestopalov

**Affiliations:** Nikolaev Institute of Inorganic Chemistry SB RAS, 3 Acad. Lavrentiev ave., 630090 Novosibirsk, Russia; vorotnikova@niic.nsc.ru

**Keywords:** bioimaging, lanthanide, octahedral halide cluster complex, photodynamic therapy (PDT), quantum dot, red/near-infrared emission, ruthenium complex, sensors, silica, upconversion nanoparticles (UCNP)

## Abstract

The low absorption of biological substances and living tissues in the red/near-infrared region (therapeutic window) makes luminophores emitting in the range of ~650–1350 nm favorable for in vitro and in vivo imaging. In contrast to commonly used organic dyes, inorganic red/NIR emitters, including ruthenium complexes, quantum dots, lanthanide compounds, and octahedral cluster complexes of molybdenum and tungsten, not only exhibit excellent emission in the desired region but also possess additional functional properties, such as photosensitization of the singlet oxygen generation process, upconversion luminescence, photoactivated effects, and so on. However, despite their outstanding functional applicability, they share the same drawback—instability in aqueous media under physiological conditions, especially without additional modifications. One of the most effective and thus widely used types of modification is incorporation into silica, which is (1) easy to obtain, (2) biocompatible, and (3) non-toxic. In addition, the variety of morphological characteristics, along with simple surface modification, provides room for creativity in the development of various multifunctional diagnostic/therapeutic platforms. In this review, we have highlighted biomedical applications of silica-based materials containing red/NIR-emitting compounds.

## 1. Introduction

Luminescent bioimaging has become one of the most widely used techniques to study the processes occurring at the cellular and molecular levels, both in vitro and in vivo [1,2,3,4,5]. As the name implies, the method itself requires the utilization of substances possessing bright emission when excited by light. Considering the therapeutic window (also known as a near-infrared window) in biological tissues, it is convenient to use compounds emitting in the range of ~650–1350 nm, i.e., in the red and near-infrared regions of the spectrum, because (1) photons in these regions mostly not absorbed by tissues and, therefore, can penetrate deep into tissues, allowing non-invasive detection of biological molecules or events; (2) since the autofluorescence of living tissues and cells is observed primarily in the violet to blue range, imaging in the IR region also provides an advantage in detection sensitivity due to the improved signal-to-noise ratio; (3) an additional advantage of red-NIR photons is their limited scattering compared to visible light photons, resulting in improved image resolution [6]. Also, some emitting compounds are able to respond to certain components or conditions of the environment, allowing them to serve as sensors for various external factors, such as temperature (intracellular luminescent thermometers), oxygen (oxygen levels inside living systems) or other gases, the presence of certain molecules (proteins, amino acids, antibiotics, DNA molecules, etc.) or ions (bio- and immunosensors) in solution or a living system. Photosensitizers (PS) are specific luminophores able to transfer the absorbed energy to molecular triplet oxygen (^3^O_2_ (^3^Σ^−^_g_)), transforming it into the excited singlet state (^1^O_2_(^1^Δ_g_)). Singlet oxygen is a type of reactive oxygen species (ROS), and its generation inside cells leads to irreversible and irreparable oxidative destruction of membranes and organelles and, ultimately, cell death. This effect is exploited in photodynamic therapy (PDT), one of the most promising cancer treatment methods. Thus, we can conclude that luminescent compounds have a wide range of applications in biology and medicine, which, combined with the active development of these fields, has led to an enormous growth of publications on the design and synthesis of luminophores promising for bioimaging [4,5], biosensing [7,8,9], diagnosis [1,3,10], and therapy [10,11,12].

Today, organic dyes are undoubtedly the most widely used luminophores in biomedicine [13,14,15]. However, there are also promising inorganic compounds, including ruthenium complexes [16,17], quantum dots (QD) [18,19,20,21], lanthanide compounds [22,23,24,25], and some others [26,27,28,29,30,31,32,33,34]. Separately, we can highlight transition metal cluster compounds, which are relatively less studied but promising for biomedical applications [35]. Despite the various advantages of each of the aforementioned compounds, all of them also have serious disadvantages, especially when used in an individual form. For example, despite high absorption coefficients, quantum yields, and bright emission, organic luminophores exhibit low photostability, i.e., irreversible destruction under irradiation and high emission sensitivity to dye concentration (self-quenching) [36]. Inorganic luminophores often do not have these disadvantages, but they also have several serious problems. For example, ruthenium and lanthanide complexes share the same problem—they are often insoluble and/or unstable in biological media due to the coordination of hydrophobic (poly)aromatic ligands (antennas), which are typically used to enhance absorption and luminescence [37,38,39]. Quantum dots are also often hydrophobic due to long aliphatic molecules coordinated to the surface to stabilize them and prevent agglomeration [40]. In addition, the most studied QDs—CdQ (Q = S, Se, Te)—can release highly toxic Cd^2+^ ions into the environment [41]. In turn, the main drawback of transition metal cluster complexes is the insolubility of the majority of them in water and their low hydrolytic stability [35].

Nevertheless, scientists have found many ways to improve the stability and biocompatibility of the entitled compounds. One of the most common methods of stabilizing luminophores is their incorporation into a specific matrix, e.g., organic polymers [42,43], inorganic oxides (SiO_2_, TiO_2_, ZnO, etc.) [44,45], metal-organic frameworks (MOF) [46,47], etc., which protects the unstable compound from the external environment. In addition to increasing stability, proper matrix selection can enhance desired properties, such as luminescence, sensing, or photosensitization and the appearance of additional matrix-derived properties. Moreover, simple and well-established methods have been developed for modifying most matrices to provide additional modalities such as targeting, specific interactions, pH-sensitive cleavage, etc. In this way, the creation of luminophore-matrix composite materials greatly expands the range of possible applications for the known luminescent compounds.

This review aims to highlight the materials based on red/NIR-emitting inorganic compounds (lanthanide complexes and upconversion nanoparticles (UCNPs), quantum dots of different nature, ruthenium complexes, and transition metal cluster complexes) and one of the most widely used matrices—SiO_2_, and to demonstrate their possible applications in biological and medical areas (see Graphical Abstract). Of course, the pool of red/NIR emitters is not limited to these compounds, but their choice is due to the significant development of the biomedical field. We believe that this paper will be helpful for both industrialists and academic researchers working in materials chemistry, biochemistry, biomedicine, etc.

## 2. Amorphous Silicon Dioxide (Silica, SiO_2_)

The development of materials based on silica is probably one of the most developed areas in biomedical applications [48,49,50,51]. Such high level of attention is due to a number of interesting properties of amorphous silicon dioxide: (1) SiO_2_ possesses high chemical inertness and thermal stability, making it an ideal matrix for materials development; (2) SiO_2_ particles have high colloidal stability resulting from high electrostatic repulsion caused by negative surface charge at neutral or alkaline pH (isoelectric point at pH = ~2–3) [52]; (3) SiO_2_ is transparent to light (E_g_ ~ 9 eV [53,54]) and magnetism—it does not absorb light or interfere with magnetic fields, allowing dopants with different properties to retain their optical and/or magnetic properties when combined in silica; (4) the surface of SiO_2_ is hydrophilic and silica is considered to be non-toxic and biocompatible, making it suitable for biomedical applications [48,49,50,51]; (5) the methods of preparation and modification of SiO_2_ are well developed and relatively simple [55,56,57], which undoubtedly increases the interest in scientists to silica-based materials.

There are two main approaches to obtaining solid (i.e., non-porous) SiO_2_ particles or silica-based materials—the Stöber process and the microemulsion method (reverse or water-in-oil (W/O) microemulsion method). Both approaches are based on the hydrolysis of alkoxysilanes, mainly tetraethoxysilane (TEOS)—Si(OEt)_4_. The first method, consisting of ammonia hydrolysis of TEOS in an ethanol-water mixture, was proposed in 1968 by Werner Stöber and his team [58]. This method is quite convenient due to its simplicity and the possibility of replacing (limited) ethanol with other solvents, depending on the nature of the dopant to be incorporated. However, the main drawback of the Stöber process is the difficulty in obtaining monodisperse particles with diameters smaller than 100 nm. A more versatile approach is the microemulsion method, which uses a non-polar water-immiscible solvent as the main phase of the colloidal system and surfactants [59,60]. In this case, the water in which the hydrolysis of TEOS takes place is entrapped in reverse micelles formed by surfactants, which act as templates for the particles formed. Such micelles have a very narrow distribution (usually smaller than 100 nm), which can be easily controlled by regulating the system components ratio.

Methods for the preparation of mesoporous SiO_2_ particles (mSiO_2_), which are particularly interesting from the point of view of biology and medicine [51], should be highlighted separately. Typically, mSiO_2_ is prepared using the slightly modified approaches described above, namely, by introducing into a system an additional surfactant (usually cetyltrimethylammonium bromide, CTAB), which forms rod-shaped ordered micelles that act as internal templates during alkoxysilane hydrolysis [51,61]. The surfactant can then be easily removed by simply washing with a solvent of specific pH. As a result, silica particles with an ordered porous structure are formed. The high interest in mSiO_2_ is due to the possibility of incorporating many bioactive molecules, such as chemotherapeutic drugs, into its pores [51]. Combined with the simplicity of surface modification with functional molecules, e.g., pH-sensitive or targeting, as well as the possibility of impregnation with visualization agents such as luminophores (luminescence imaging), electron-dense particles (computed tomography), or MR contrast compounds (magnetic resonance imaging, MRI), this makes mesoporous silica particles an excellent platform for the development of drugs with therapeutic and diagnostic effects (theranostic agents) [51].

Further, each section of the review will be structured according to the following logic: (1) general information about the luminophore, (2) classification of the main methods of obtaining and applications of silica-based materials in biomedicine, (3) discussion on several most representative articles in the field (according to the authors’ opinion).

## 3. Lanthanides (Ln) and Upconversion Nanoparticles (UCNP)

The lanthanide (Ln) group includes fifteen elements from lanthanum to lutetium. All lanthanides in the Ln^3+^ form, except La^3+^, Pm^3+^, and Lu^3+^, demonstrate bright emission [62]. The luminescent properties of lanthanides are characterized by narrow emission bands, photostability, and long luminescence lifetimes. As can be seen from Figure 1A, lanthanide luminescence covers a huge spectral range from the ultraviolet to the near-infrared [63]. The maxima of these bands, unlike the intensity, do not alter significantly by the changes in ligand environment or parameters such as temperature, pressure, or pH. One of the main disadvantages of these compounds is their low absorption coefficient, resulting in low luminescence intensity. This problem was, to some extent, solved in the middle of the 20th century with the discovery of the “antenna effect” [62]. The effect is based on antenna ligands (usually aromatic or polyaromatic molecules) capable of enhancing metal-centered emission by transferring energy from the ligand to the metal. This discovery aroused great interest in lanthanide complexes and served as an impetus for the development of the area. It is important to note another prominent property of lanthanide luminescence, namely, the possibility of excitation by two or more low-energy photons (IR) with emission of photons of higher energy (e.g., UV, visible region, or IR) (Figure 1B). This effect is called upconversion luminescence (UCL) [62,64].

Currently, lanthanides are used in various practical applications, including catalysts, optoelectronic components, magnets, lasers, and many others [62]. Naturally, the rich luminescent properties of lanthanides are also attracting interest from the point of view of biology and medicine. Of particular interest to these fields are those lanthanides that emit in the red and infrared regions of the spectrum. Concerning the classical type of luminescence, in which a photon of higher energy is absorbed and a photon of lower energy is emitted, most studies have focused on europium (Eu^3+^) complexes [65]. At the same time, we should not forget about the ability of lanthanide compounds to UCL, which allows both excitation and emission in the IR region.

It should be noted that the UCL of lanthanide molecular complexes is rather inefficient, and nowadays, nanoparticles of a certain structure doped with trivalent lanthanide ions (upconversion nanoparticles—UCNPs) are used as UC luminophores [62,64]. In these systems, crystalline metal fluoride nanoparticles such as LaF_3_, YF_3_, NaYF_4_, BaYF_5_ and others are typically used as the matrix, Yb^3+^ and Nd^3+^ ions are usually used as dopants, while Er^3+^, Tm^3+^, and Ho^3+^ are also added to enhance absorption in the IR region. In addition, Gd^3+^, currently considered one of the best MR contrast agents, is often used as a co-dopant to enhance the modality of the system (i.e., to impart an additional property). These systems thus can combine several properties that allow their visualization in living systems in multiple ways—luminescence in the IR region, MR contrast, and X-ray contrast (due to the high molar mass of lanthanides and their high local concentration). Despite the great prospects, UCNPs are highly unstable in aqueous media, and for their stabilization, researchers actively exploit surface coating with silica.

### Ln-Containing Silica-Based Materials

Methods for the preparation of Ln-containing silica-based materials can be primarily divided into two main groups, in which either UCNPs or lanthanide molecular complexes are used. The first group includes coating pre-synthesized lanthanide nanoparticles with silicon dioxide (microemulsion, Stöber process, or pyrolysis) [66,67,68,69,70,71,72,73,74,75,76,77,78,79,80,81,82,83,84,85,86,87,88,89,90,91,92,93,94,95,96,97,98,99]. The second group can be further subdivided into three groups, namely: (1) coating of pre-synthesized SiO_2_ particles with groups able to bind metal ions via covalent bonds followed by soaking with lanthanide simple salts/molecular complexes [100,101,102,103,104,105,106,107,108,109,110,111]; (2) utilization of lanthanide complexes containing ligands modified with alkoxysilane groups in the nanoparticle preparation process [111,112,113,114,115,116,117,118]; (3) hydrolysis of alkoxysilanes in the presence of lanthanide simple salts/molecular complexes containing no functionalized ligands, i.e., impregnation without any bonds formation [119,120,121,122,123,124,125,126,127]. Based on the quantity and quality of publications, it can be concluded that the use of UCNPs is the most popular approach among researchers. Since the early 2000s, a large number of papers have been published on the study of this topic, in which the authors mainly investigated the fundamental properties of such materials, gradually focusing the attention of readers on the prospects of biomedical applications [66,67,68,69,70,71,72,100,101,102,103,117,118,119,120,121]. A small number of papers have also been published demonstrating the possibility of using the materials as sensors for temperature or certain molecules [90,91,93,104,115,122] and for the detection of certain types of microorganisms [74,105,115,116,122]. Since 2012, the active study of such materials in living objects has begun, as evidenced by the explosive growth of publication activity. In these studies, Ln-containing materials have been considered as imaging agents in vitro [74,75,76,77,78,79,80,81,82,83,84,85,92,93,94,106,107,108,109,110,111,112,113,126,127,128] and in vivo [75,76,77,78,79,80,81,93,94,95,111,113,126,127,128], as well as platforms for the development of systems for targeted cancer therapy. These systems use special blocking molecules or particles that detach under certain conditions (IR radiation causing lanthanide emission, change in pH, etc.), thereby causing the release of the drug [80,81,82,86,93,124,125,126,128]. There are also a small number of papers demonstrating the efficacy of such delivery systems in the treatment of mouse-bearing grafted tumor [76,81,86,93,96,124,126,128].

One of the first significant studies on the biological activity of Ln-containing SiO_2_ materials is the study of Sun and co-workers [79], where the authors combined upconversion luminescence of UCNPs NaYF_4_:Yb, Tm@NaGdF_4_ and classic (down-conversion) luminescence of lanthanide complexes. The material was obtained in several stages: (1) preparation of NaYF_4_:Yb, Tm@NaGdF_4_; (2) covering of the NPs surface with mesoporous SiO_2_ in the presence of CTAB in a mixture of cyclohexane/water; (3) surface modification with the silane containing dibenzoylmethane (dbm); (4) soaking of dbm-modified particles in the solution of Ln(dbm)_3_(H_2_O)_2_ (Ln = Eu, Sm, Er, Yb, and Nd). The average size of the particles was determined to be 95–110 nm (Figure 2A). The system obtained demonstrates classical red/NIR emission of the Ln(dbm)_4_ complexes on the surface in the 600–1700 nm range depending on the Ln used and upconversion emission of the core at 800 nm. Biological studies were conducted using the UCNPs@mSiO_2_-Eu(dbm)_4_ system (λ_em_ = 613 and 800 nm). The cytotoxicity study on HeLa cells (human cervical cancer) showed that nanoparticles have no effect on the viability of cells up to a concentration of 400 µg mL^−1^ (Figure 2B). According to CLSM (confocal laser scanning microscopy) (λ_ex_ = 405 nm, λ_em_ = 600–700 nm), the particles penetrate the cells and localize within the cytosol (Figure 2C). To study in vivo imaging, the dispersion of the particles (100 μL, 1 mg mL^−1^) was injected in the tail vein of a nude mouse. After 1 h since administration, a strong luminescent signal was observed (λ_ex_ = 980 nm, λ_em_ = 800 ± 12 nm) (Figure 2D). By studying ex vivo emission, it was shown that these particles are mainly distributed in the liver and lung.

In addition to bioimaging, UCNPs may also be of interest for creating systems for targeted drug delivery and controlled drug release, as demonstrated by Chien and co-workers [76]. The structure of the system created by the authors can be divided into four levels, each with a different purpose: (1) the NaYF_4_: Yb/Tm (UCL λ_ex_ = 980 nm, λ_em_ = 360, 480, 800 nm); (2) SiO_2_ layer (microemulsion method, d ~ 50 nm (Figure 3A)) modified with -NH_2_ groups and doxorubicin (DOX) molecules (cytostatic drug); (3) PEG-modified with folic acid (FA) (targeted delivery); (4) 2-nitrobenzylamine molecules bound to folic acid entrapping DOX inside the system (photoinduced release of DOX). Thus, after targeted delivery to the tumor site, the material is irradiated with NIR light. In turn, emitted UV light (λ_em_ = 360 nm) induces photooxidation and cleavage of 2-nitrobenzylamine molecules, resulting in the release of the DOX. The possibility of detecting NIR luminescence of NaYF_4_:Yb/Tm (λ_ex_ = 980 nm, λ_em_ = 800 nm) inside the organism is an additional diagnostic tool. The material penetrates well into the cells locating in the cytosol (Figure 3B) and exhibits low toxicity without irradiation and high toxicity under NIR laser irradiation. The anticancer activity was studied in mice with grafted tumor (HeLa). Using in vivo imaging, it was shown that after injection of the sample into the tail vein, no emission was observed. Nevertheless, after irradiation of the tumor site with the NIR laser, NIR emission is observed after 1 h of administration, indicating the release of DOX (Figure 3C). Such treatment of a grafted tumor resulted in the complete prevention of tumor growth for nine days.

In [126], the authors used a slightly different approach. Instead of UCNP@SiO_2_ composites, the authors used mesoporous silica nanoparticles (Stöber process, d~110 nm) doped with Eu^3+^ (luminescent imaging) and Gd^3+^ ions (MR contrast). To incorporate the anticancer drug (camptothecin—CPT), the authors modified the SiO_2_ surface with thiol groups (-SH), which can form disulfide -S-S- bonds with CPT. This type of bond is able to be slowly reduced by glutathione in the body, thereby releasing the drug. The surface was additionally modified with folic acid to impart targeted delivery properties. The authors showed that the materials penetrate the cells, while modification with FA increases the penetration rate (Figure 4A). According to cytotoxicity studies, CPT-containing particles, unlike pure material, demonstrate high toxicity against L929 (*Mus musculus* fibroblast cell line) and HeLa cells, while modification with FA additionally increases the toxicity due to target delivery (Figure 4B). In vivo, photoluminescent imaging after intravenous administration of the material in mice bearing grafted tumor demonstrated bright red emission in the tumor site (Figure 4C). The treatment with FA-modified europium-containing mSiO_2_ resulted in almost complete disappearance of the tumor after 23 days.

In [94] UCNPs NaYF_4_:Yb^3+^, Er^3+^@NaYF_4_ (imaging and luminescent thermometer), commercial photosensitizer Chlorin e6 (Ce6), and citrate-stabilized CuS nanoparticles as agents for photothermal therapy (PTT) were combined within one system. The system was synthesized as follows: (1) preparation of NaYF_4_:Yb^3+^, Er^3+^@NaYF_4_ nanoparticles; (2) synthesis of citrate-stabilized CuS nanoparticles (d = 12 ± 1.2 nm); (3) coating of UCNPs with Ce6 modified layer of mesoporous silica (Stöber process, d = 89.4 ± 7.6 nm); (4) soaking of the UCNPs-Ce_6_@mSiO_2_ in the dispersion of CuS (Figure 5A). The cytotoxicity of the materials was studied on MCF-7 (human breast cancer) and B16 (murine melanoma) cell lines. According to the data obtained, the materials show low toxicity in the dark and high toxicity under irradiation with IR laser (λ = 980 nm). The cytotoxicity was significantly higher for the CuS-modified sample. According to CLSM, UCNPs-Ce_6_@mSiO_2_-CuS shows good penetration into both types of cells, and green and red emission channels can be used for cell imaging (Figure 5B). The anticancer efficacy was investigated in a mouse-bearing grafted tumor (B16). After sample injection and irradiation with IR laser for 5 min, tumor growth was monitored for 11 days. It can be seen in Figure 5C that systems modified only with Ce_6_ or CuS showed lower efficiency compared to the system containing both therapeutically active components.

## 4. Quantum Dots (QD)

Quantum dots (QDs) are semiconductor crystalline nanoparticles ranging in size from 1 to 10 nm (e.g., InGaAs, CdQ (S, Se, Te), or GaInP/InP) coated with a monolayer of organic stabilizer molecules. Due to such small size, the electronic and optical properties of these materials are strongly influenced by the quantum size effect, and, as a consequence, QDs occupy an intermediate position between bulk semiconductors and molecular compounds. In other words, the properties of quantum dots are directly dependent on their physical size, i.e., a change in their size leads to a change in the width of the energy gap and, hence, in the emission wavelength or the conductivity (Figure 6) [129].

The outstanding narrow-band emission, combined with the possibility to precisely tune their position, has led to a growing interest in the scientific community in this class of compounds. Since then, scientists have developed and optimized many synthetic methods for obtaining QDs of a given size in large quantities [129,130]. Currently, quantum dots are widely used as transistors, light-emitting diodes, components of solar cells, and diode lasers [131]. They are also of interest as an alternative to organic phosphors in bioimaging [132,133,134].

One of the major problems of quantum dots application as phosphors is their relatively low quantum yield, which is associated with the non-radiative recombination of the electron-hole pairs at the surface of the particle. This problem was solved by creating particles of the core-shell type (the “core” of one semiconductor is covered by the “shell” of another one). There are two types of core-shell QDs (Type I and Type II) according to the nature of the electronic structures of the semiconductors used (Figure 7) [135]. Type I QDs are those in which a wider bandgap semiconductor is used as the shell, which acts as a passivator of surface states and localizes the electron-hole pair inside the core. This approach significantly increases luminescence efficiency. Type II dots are particles in which the band gap values of the semiconductors are comparable, but band potentials are shifted relative to each other, which allows the localize charge carriers (electrons and holes) in different parts of the nanocrystal—electrons in the shell, holes in the core (or vice versa). This approach, due to the spatial separation of carriers, delays the recombination rate and allows us to obtain systems with long lifetimes. Also, in this case, a shift of the emission maximum to the red region (up to the IR region) is observed.

Thus, as mentioned above, tuning the luminescence of QDs by varying their size and composition, i.e., obtaining compounds with well-defined optical properties and small particle size, makes quantum dots promising for application as luminescent biomarkers and agents for in vitro and in vivo imaging. The most common compounds in this field are cadmium chalcogenide quantum dots (CdQ, where Q = S, Se, or Te) due to their high photophysical characteristics (λ_em_ varies from 500 to 800 nm), relative ease of preparation, and well-studied methods for further modification. The main approach for their preparation is colloidal synthesis, in which nanocrystal growth occurs in high-boiling solvents at high temperatures in the presence of stabilizing molecules (usually a mixture of trioctylphosphine (TOP) and trioctylphosphinoxide (TOPO) acts as solvent and stabilizer simultaneously). Since the particle size in this method is a function of the reaction time, it is quite easy to isolate particles of the desired size by stopping the reaction (cooling) [130,132]. The main drawbacks of cadmium QDs include the primary hydrophobic coating and slow degradation of the nanocrystals in aqueous solutions accompanied by the release of toxic metal ions. Currently, these problems are successfully addressed by substituting the hydrophobic coating with a hydrophilic one and/or incorporating the nanocrystals into insoluble inert matrices such as silica.

### QD-Containing Silica-Based Materials

In the case of QD-containing silica-based materials, due to the possibility of replacing the primary stabilizing layer of quantum dots, four methods of coating with silicon dioxide can be distinguished: (1) microemulsion method using hydrophilic silanes (such as TEOS, 3-aminopropyltriethoxysilane (APTES), etc.) and hydrophobic quantum dots (usually using unmodified QDs, i.e., those coated with TOPO/TOP monolayer) [136,137,138,139,140,141,142,143,144,145]; (2) preparation of hydrophilic QDs either in situ or by substitution of hydrophobic surface groups followed by silica coating via the Stöber process or microemulsion method [146,147,148,149,150,151,152,153,154,155]; (3) sequential coating of hydrophobic QDs with a series of silanes in the following order: hydrophobic—amphiphilic—hydrophilic [156,157]; (4) pre-substitution of the QD surface coating with silane-containing groups and subsequent co-hydrolysis with another silane [158,159]. Currently, most of the studies devoted to these materials aim to investigate the fundamental properties of such systems [142,143,144,145,146,147,158,160]. Regarding biomedical applications, there are many papers devoted to the study of cellular toxicity and in vitro bioimaging [137,138,139,140,141,148,149,150,151,152,153,154,155,156,159,161,162], biological effects in vivo in the mouse model—acute toxicity and bioimaging [140,141,150,151,152,153,154,155,157,162,163,164] as well as the development of the systems for cancer therapy [140,150,154].

One of the first papers studying in vivo luminescence imaging using silica-coated quantum dots is [152]. The authors used an unusual morphology of quantum dots—rod-shaped QDs or quantum rods (QRs). CdSe/CdS/ZnS QRs were obtained in two steps: (1) synthesis of CdSe QRs by interaction of cadmium oxide with TOP-Se in the presence of tetradecylphosphonic acid and TOPO at 300 °C; (2) coating of CdSe QRs with CdS/ZnS layer by their interaction with cadmium oxide, zinc acetate, and TOP-S in oleic acid solution at ∼210 °C. Coating the obtained particles with a silica layer of different thicknesses was performed by hydrolysis of TEOS using the microemulsion method. The authors studied the luminescence properties of the particles as a function of silica layer thickness. They showed that increasing the thickness leads to an increase in emission intensity and quantum yield, indicating better shielding from the environment. However, a decrease in the colloidal stability of the system in aqueous medium and under physiological conditions was observed with the shells thicker than 25 nm. Thus, the most suitable were SiO_2_-coated QRs with a shell thickness of ~10 nm (Figure 8A). The cytotoxicity study showed that the nanoparticles had almost no effect on the viability of the Panc 1 (human pancreatic cells) and RAW 264.7 (macrophage) cell lines up to 20 μg mL^−1^. The CLSM (λ_ex_ = 442 nm) showed that the particles penetrate well into the cells (Figure 8B), and the penetration mechanism is phagocytosis. Since the obtained system showed bright emission, high colloidal stability, low cellular toxicity, and good cellular penetration, the authors decided to investigate the possibility of its application for in vivo luminescence imaging of tumors. Therefore, the material with a silica shell thickness of ~10 nm was injected into the tumor site of a mouse-bearing grafted tumor. The body of the mouse was imaged using a light source with a wavelength of 488 nm. The results showed that 5 min after injection, a rather strong luminescence signal was detected at the tumor site, while after 10 min, the particles were evenly distributed throughout the tumor volume (Figure 8C), confirming this system to be promising for in vivo imaging.

In [157], different materials containing quantum dots were studied for in vivo luminescent imaging. Type II cadmium quantum dots—CdSe/ZnS—were chosen as luminophores. To stabilize QDs in aqueous medium and make them biocompatible, the authors proposed two approaches: (1) coating of QDs with PAMAM-C12 dendrimer (polyamidoamine); (2) coating of dendrimer-stabilized QDs with silicon dioxide and subsequent modification of particle surface with amino groups, PEG1100, and DOTA-phenyl-isothiocyanate-Gd (Figure 9A). It was shown that intravenous injection of the dendrimer-stabilized dots (the first approach) into the tail vein of mice caused a decrease in blood pressure and heart rate, which appears to be due to the degradation of the system in the body and the release of PAMAM and QD in an individual form. By cross-linking the dendrimers, the authors successfully increased the stability of the system and reduced the overall toxic effect. At the same time, experiments on the visualization of mouse brain vessels showed that the necessary resolution could not be achieved with either drug. The second approach used was more successful. However, the authors note that the particles cannot be used in vivo in individual form due to their low colloidal stability under physiological conditions. To overcome this problem, the particles were covalently coated with PEG1100. The obtained system showed no adverse effects on the organism of experimental animals during intravenous administration. In addition, this system showed good results in brain vascular imaging experiments in contrast to the system with PAMAM. Finally, the authors demonstrated the possibility of luminescent imaging in vivo in a mouse-bearing grafted cancer. For this purpose, an intravenous injection of particle suspension (1.6 nmol kg^−1^) was performed, followed by imaging of the whole body of the mouse (λ_ex_ = 450 ± 30 nm, λ_em_ = 650 nm) (Figure 9B). The obtained data indicated that 90 min after injection, the maximum emission intensity was observed in the liver region. At the same time, the emission intensity in the tumor was three times higher than in the rest of the body, confirming the prospect of using such a system for in vivo imaging but undoubtedly requiring further development.

Another type of red-emitting quantum dots are carbon QDs or CDs. In [155], silica-coated CDs were used for targeted fluorescence imaging of cervical cancer by recognising the epidermal growth factor receptor (EGFR). The material preparation route consisted of several steps: (1) synthesis of CDs via thermal decomposition of citric acid and urea in DMF (N,N-dimethylformamide) at 180 °C; (2) coating of CDs with the silica layer via microemulsion method; (3) coating of CDs containing silica particles with polyacrylamide layer via reverse microemulsion polymerization of acrylamide (Aam, monomer) and N,N′-methylenebisacrylamide (BIS, cross-linker) in the presence of the N-terminal nonapeptide of EGFR modified by palmitic acid at the C-terminal. During the polymerization process, the terminal hydrophobic domain of the peptide is located outside of the micelles, while the hydrophilic part prefers aqueous media inside the micelles. The polymerization and subsequent elution of EGRP results in the formation of an epitope-imprinted polymer (C-MIP) layer capable of recognizing specific molecules (in this case, EGRP), while silica-coated carbon QDs were used as luminophores. The diameter of C-MIP was calculated to be ~60–65 nm (Figure 10A). The cytotoxicity study showed that the nanoparticles have almost no effect on the viability of HeLa and MCF-7 cell lines up to 500 μg mL^−1^. According to CLSM, the particles accumulate better in HeLa cells (Figure 10B) than in MCF-7 cells due to the higher EGFR expression in the former cell line. Overall, this confirms the recognition of the receptor by C-MIP. In vivo, imaging was conducted in mice bearing grafted HeLa tumor. After intraperitoneal injection of the materials, maximum emission was observed at 22 h post-injection and decreased gradually over time (Figure 10C).

It should be noted that some types of quantum dots can photosensitize the singlet oxygen generation process, allowing them to be used in PDT. In a series of investigations conducted in the group of Prof. D. Yang [150,151,153,154], a multifunctional targeted/therapeutic system based on mSiO_2_ containing graphene QDs (GQDs) was developed. The authors showed that after loading with DOX and further surface modifications, such a system could be used for pH-induced [150] or targeted [151,153] chemotherapy. In the recent paper [154], graphene QDs were used as PSs for PDT acting upon two-photon excitation (635 nm), while DOX was added to the system for additional effect and released at acidic pH. The resulting particles with a diameter of ~135 nm (Figure 11A) showed low toxicity in the dark and noticeable phototoxicity upon laser irradiation. Impregnation of the system with DOX further enhanced both types of toxicity. The anticancer efficacy was tested in nude mice bearing grafted HeLa tumor by intravenous administration of the material followed by irradiation with laser (635 nm) for 10 min (Figure 11B). The data obtained showed a significant decrease in tumor growth rate, confirming the efficiency of the developed system in combined PDT/chemotherapy. According to histological analysis, the red fluorescence of CQDs was observed in cancer tissues, indicating the high permeability and retention effect of nanomaterials in solid tumors (Figure 11C).

## 5. Ruthenium Complexes

The study of ruthenium complexes and the possibility of their application in biology and medicine have been actively developed over the last few decades [165,166,167]. Such active development in the field is explained by the need to search for new types of anticancer drugs, driven by the rapid increase in cancer patients and high mortality [168,169]. Researchers have found that ruthenium complexes possess pronounced cytostatic effects similar to platinum compounds while having generally lower adverse effects [170,171]. Another interesting fact is the possibility of intracellular reduction of inactive Ru(III) complexes to cytostatic Ru(II) ones, allowing the introduction of primarily non-toxic compounds. These facts have significantly increased the interest of researchers in such compounds, due to which thousands of ruthenium complexes of various compositions are now known, and their biological effects are well enough studied [165,166,167]. The first significant achievement in the development of drugs based on ruthenium complexes was the clinical trial of the compound NAMI-A ([ImzH][*trans*-RuCl_4_(DMSO)(Imz)], Imz—imidazole), which, however, failed due to low therapeutic efficacy and ambiguity of the data. Nevertheless, two other Ru-based drugs, namely BOLD-100 (Na[*trans*-RuCl_4_(Indz)_2_]) and TLD1433 (Ruvidar, Ru(II) polypyridyl complex incorporating an oligothienyl-containing ligand) have passed I phase of clinical trials and showed high efficacy. Now, they are passing phase II (NCT04421820 and NCT03945162 respectively).

In addition to their high potential as cytostatic agents, ruthenium complexes exhibit bright red emission with high lifetime and quantum yield values and the ability to photosensitize the singlet oxygen generation process. Moreover, some Ru complexes, similar to UCNPs, can be excited by the high-efficiency absorption of two low-energy photons. All this undoubtedly indicates the prospect of their application in bioimaging and PDT and makes them even more interesting candidates for the role of anticancer drugs [172,173]. Over the past ten years, a large number of studies have been conducted to investigate and improve the efficacy of ruthenium complexes for PDT on model objects (cells, rodents), resulting in the accumulation of a large amount of knowledge in this area [172,173].

Another possible application of ruthenium complexes in cancer therapy is based on the photoactivated effects, the so-called photoactivated chemotherapy (PACT) [174]. These effects are indirectly related to the luminescent properties of the complexes, particularly their ability to be transferred into an excited state upon photon absorption. It opens a large number of therapeutic effects, such as: (1) photoinduced substitution of the ligand(s) by water and formation of a cytostatically active form, allowing the administration to the patient of a drug that initially has no cytostatic activity; (2) the use of ligands able to bind to DNA, which allows its further photooxidation causing irreversible destruction of the molecule; (3) binding of chemotherapeutically active compounds to a ligand or metal center and their subsequent photoactivated cleavage, etc. This area is currently poorly studied, and the number of related publications is rather small, but further development is expected.

Thus, the research on Ru complexes related to their application in medicine has already achieved significant results. This is confirmed by many publications on this subject and clinical trials [175,176]. However, despite the less toxic effects compared to Pt(II) complexes, these compounds share a common drawback—the lack of targeted delivery in individual form. In general, worldwide studies show that the incorporation of active components into various matrices is not only the most convenient method to impart target delivery property, but such an approach also results in a more biocompatible and stable system. This approach is actively exploited in the case of ruthenium complexes. There are a lot of papers devoted to the incorporation of complexes into silica, which will be discussed in detail in the following section.

### Ruthenium Complexes-Containing Silica-Based Materials

Generally, the methods of obtaining such materials can be divided into two groups. The first approach is the simplest and, therefore, the most popular—the incorporation of complexes into the silica without the formation of covalent bonds with the matrix [177,178,179,180,181,182,183,184,185,186,187,188,189,190,191,192,193,194,195,196,197,198]. In this case, the complex can be sorbed by pre-synthesized mSiO_2_ or solid silica particles can be obtained in the presence of the complex. The second approach is aimed at the formation of covalent bonds between the complex and the matrix. It can be divided into two subgroups: (a) modification of the ligand with a silanol group to form [RuL’_2_L-Si(OR)_3_]complex [199,200,201,202,203,204,205]. The silanol group can hydrolyze similarly to TEOS with the formation of a RuL’_2_L-Si-(O-Si-) bond type. In this case, coating of pre-synthesized particles and simultaneous hydrolysis of the complex and silane are possible; (b) surface modification of silica particles with groups able to bind ruthenium complexes [206,207,208,209,210,211,212,213].

Due to the wide variety of properties of the initial compounds, the areas of application of the materials are also very diverse. In addition to the study of fundamental properties [197,198,199,200,201], these systems are considered sensors (including intracellular) for various external conditions such as oxygen content [191,192,193,204,205,206], other molecules or ions [190], pH of the environment [185], and temperature [187,188,189], agents for bioimaging [183,184,185,186,188,202,205], immunosensors and detectors for biomolecules and cells of a particular type [177,178,179,180,181,201,202,203,213], as well as systems for cell imaging and/or for delivery of cytostatic/photosensitizing ruthenium complexes or other molecules into the cell [185,194,195,196,207,208,209,211,212,213].

An interesting example of the use of such systems as sensors for certain biomolecules is [178]. In this study, the authors created a system for the fluorescent quantitative detection of the HER2/neu gene, an important biomarker in diagnosing some types of cancer, such as breast cancer [214,215]. To obtain the particles, the Stöber process was used: hydrolysis of TEOS in water-ethanol mixture in the presence of [Ru(phen)_3_]Cl_2_ complex (phen is 1,10-phenanthroline). In this way, the authors obtained three types of particles with diameters of 65 ± 8 nm, 440 ± 18 nm and 800 ± 20 nm (Figure 12A). The study of the photostability of the obtained system showed a significant increase in stability due to the shielding of the complex within the matrix (Figure 12B). Next, the authors performed a surface modification of silica by hydrolysis of 3-mercaptopropyltrimethoxysilane in the presence of particles. The modified particles were then coated with streptavidin, able to pair with biotin-modified anti-HER2/neu antibody. The assay itself was as follows: the amino-group modified slide was incubated in HER2/neu solution, washed, and placed in a suspension containing anti-HER2/neu modified particles, and then washed again to remove unbound particles. The resulting glass was examined with a fluorescence microscope, and the dependence of the number of fluorescent particles was plotted as a function of concentration (Figure 12C). From the obtained dependence, the presence of HER2/neu can be easily determined qualitatively and quantitatively (detection limit is 1 ng mL^−1^). Other gene-antibody combinations can be used upon request.

Another interesting example of sensor system development is [201]. The authors obtained a detection system for *Giardia lamblia*, one of the most widespread parasitic microorganisms in the world. To obtain ruthenium-containing particles, the authors used an approach based on the covalent bond formation between a Ru complex and silicon dioxide. For this purpose, the complex [Ru(NH_2_-phen)_3_](PF_6_)_2_ (NH_2_-phen is 5-amino-1,10-phenanthroline) was obtained. The complex was further modified with a silanol group by reaction with succinic anhydride and APTES. Particles with an average diameter of 65 nm were prepared by the microemulsion method. Particles with physically encapsulated complex were used as a comparison sample. The higher photostability (Figure 13A) and resistance to leaching (Figure 13B) were observed in the case of covalent binding of the complex.

Further, the particle surface was modified with streptavidin, similar to the previous example. Detection of *Giardia lamblia* cysts was performed as follows: anti-*Giardia* antibodies, biotinylated rabbit IgG-class antibodies, and streptavidin-modified nanoparticles were added to the cyst solution. The mixture was then incubated for two days. Cysts were separated, washed, and examined under a fluorescence microscope. The images obtained (Figure 13C) show a bright red glow, indicating the effectiveness of the presented method in detecting *Giardia lamblia* cysts.

In [207], the authors elegantly demonstrated the use of photoinduced ligand exchange to obtain a light-activated drug delivery system. The material was prepared using MCM-41 mesoporous silica particles modified with benzonitrile groups able to coordinate to ruthenium ion. The complex was bound to the surface by the interaction of the modified MCM-41 particles with [Ru(terpy)(dppz)(H_2_O)](PF_6_)_2_ (terpy is 2,2′:6′,2″-terpyridine, dppz is dipyridophenazine) (Figure 14A). Upon impregnation, the coordinated water molecule was released, and the nitrile group was coordinated to the metal center. The efficiency of the photoactivated release was studied as follows: the sample was placed in water and irradiated with white light (λ > 450 nm), and then the emission spectra of the suspension were recorded. A gradual red coloration of the solution and a decrease in the emission intensity (caused by the formation of the aqua complex) were observed, indicating the photoactivated release of the complex. The complete disappearance of the emission was observed after 1 h of irradiation. The success of this experiment allowed the authors to proceed to the next step—to study the efficiency of binding of DNA molecules during photoactivation. Therefore, the sample was dispersed in a buffer solution (pH = 7.0), irradiated with light until the emission disappeared, and then the DNA solution was added to the system. Simultaneous appearance and enhancement of the ruthenium complex emission occurred, which indicates the ability of the complexes to bind DNA effectively. The porous nature of the matrix allows additional chemotherapeutic agents to be incorporated into the pores. In addition, ruthenium complexes on the surface of the particles can act as molecules blocking drug release from the pores, and photoactivation will result in the simultaneous release of both drugs. This can significantly reduce the negative effects of drugs on healthy tissues and increase the overall effectiveness of treatment. In this study, the authors chose the commercial cytostatic drug Paclitaxel^®^. It was shown that irradiation of doped particles does release both Paclitaxel^®^ and ruthenium complexes, but the release of Paclitaxel^®^ is somewhat delayed and occurs only after ~1 h. The final stage of the research was to study the biological effects of the system in vitro on MDA-MB-231 and MDA-MB-468 breast cancer cells. According to CLSM, the particles penetrate cells and localize throughout their volume (Figure 14B). A significant increase in the emission was observed after irradiation of the cells, indicating the binding of DNA molecules. Cytotoxicity studies in cancer cell cultures showed that the particles exhibit relatively low toxicity before irradiation, while their toxicity increases after irradiation, becoming comparable to that of the free drug (Figure 14C). All of the above results indicate a high efficacy of the developed model system for photoactivated cancer therapy.

In [195] the authors compared the efficiency of unmodified vs. amino-modified mesoporous silica as a carrier for pH-induced release of C,N-cyclometalated Ru(II) anticancer agent—[Ru(ppy-CHO)(phen)_2_]^+^ (ppy-CHO is deprotonated 4-(pyridin-2-yl)benzaldehyde) (Figure 15A). According to the data obtained, silica modification does not affect the sorption efficiency (~1.5–2% *w*/*w*), while the amino-modified material showed a much higher drug release rate at acidic pH = 5.4. Cytotoxicity studies were performed in U87 glioblastoma cells, A2780 ovarian cancer cells, and Chinese hamster ovary normal cells. The amino-modified material showed higher activity in cancer cells (in a dose-dependent manner) compared to the unmodified material and was also less toxic to normal cells. A detailed cytotoxicity study of the materials in comparison to the free complex in U87 cells revealed that ruthenium-containing amino-modified mSiO_2_ NPs have higher toxicity after both 24 and 72 h of incubation, while an 8-fold higher complex concentration is required to achieve the same activity. To study cellular uptake, fluorescein-labelled particles were obtained. CLSM shows that the use of a silica carrier results in higher intracellular accumulation (Figure 15B). Dual annexin V/propidium iodide (PI) staining was used to examine cell death after treatment with the materials and the free complex. The data obtained showed that all materials induced cell apoptosis with different efficiencies, while Ru-containing amino-modified mSiO_2_ NPs produced a greater extent of apoptotic cells (Figure 15C).

## 6. Octahedral Metal Cluster Complexes

Although “cluster complexes” or “clusters” were obtained in the middle of the 19th century, the term itself was first introduced in the papers by F.A. Cotton in 1967 [216]. Since this literature review is devoted to the application of red inorganic emitters, attention will be paid to the octahedral (six-nuclear) halide cluster complexes of molybdenum and tungsten, which possess the most outstanding luminescent properties among others of their class. Octahedral cluster complexes of molybdenum and tungsten can be represented by the general formula [{M_6_X_8_}L_6_]^n^, where M = Mo, W; X = Cl, Br, I; L is an organic or inorganic ligand, n is a charge. In general, these complexes consist of an octahedron of six metal atoms inscribed in a cube of eight so-called internal µ_3_-bridging ligands X. The resulting system {M_6_X_8_}^m^ is referred to as cluster core. In addition, each metal atom is coordinated by one apical ligand L (Figure 16).

Apical ligands strongly influence the properties of these compounds, and the relative ease of their substitution allows, to some extent, to vary or enhance the desired properties [217,218,219,220]. Clusters have a wide range of functional (i.e., having potential applications) properties, but papers demonstrating practical applications of these compounds have only been published in the last 10–15 years. One of the most interesting and multifunctional properties of the complexes is the bright luminescence in the red and near-IR spectral range (550–1000 nm) with long emission lifetimes (hundreds of microseconds) and high quantum yields (up to 88%) [217,221]. Different light sources can be used to excite cluster compounds: UV-visible light (250–500 nm) [217,218,219,221,222,223,224,225], NIR light (two-photon absorption) [226,227], X-rays [228,229,230,231,232], electron beam [233]. They are also considered to be very effective photosensitizers (Φ_Δ_ = 80–90%) [234,235]. Due to these two interconnected properties, such compounds can find applications both in the fields considered in this review, such as sensor systems, agents for bioimaging and PDT [35], and in other fields—sensors [221,231], photonics (components of lasers and waveguides) [236,237], photovoltaics (components of solar cells) [238,239], etc. However, as mentioned above, the main disadvantages of the clusters are insolubility in water and/or low hydrolytic stability. Currently, there are only four reliable examples of luminescent molybdenum and tungsten complexes relatively stable in aqueous medium, namely Na_2_[{Mo_6_I_8_}(N_3_)_6_], Na_2_[{Mo_6_I_8_}(SCN)_6_] [235], and [{M_6_I_8_}(DMSO)_6_](NO_3_)_4_ (M = Mo, W) [240,241], which, although, also tend to hydrolyze over time partially or completely. Such behavior is not conducive to the use of complexes in individual form since components of devices and sensors are often in constant contact with humid air containing water, and hydrolysis will adversely affect the performance. In the case of biological and medical applications, high stability of the compounds in aqueous environments is also required. Among others, cluster-containing silica-based materials are one of the most studied in terms of biomedical applications. However, due to the fact that this field is very young, the number of biomedically-oriented studies is still limited.

### Cluster-Containing Silica-Based Materials

The history of the materials based on cluster complexes and amorphous silica began in 2008 with a paper by Grasset and co-workers [242,243]. In these studies, the authors, for the first time, obtained silica-based materials using the microemulsion method via TEOS hydrolysis in the presence of Cs_2_[{Mo_6_X_8_}X_6_] (X = Cl, Br or I) clusters. The materials exhibited luminescent properties and dispersed well in water, making them promising for applications such as bioimaging and PDT. However, the first studies on the biological properties of this system were performed only in 2013 by Aubert and co-workers [244], which will be discussed in detail below. In general, the methods for such materials preparation can be divided into two groups: (1) hydrolysis of TEOS in the presence of the complex using the Stöber process [245,246,247] or the microemulsion method [242,243,244,245,248,249,250,251,252,253,254]. In this case, complete hydrolysis of the complex with the formation of covalent and hydrogen bonds with the matrix is observed; (2) impregnation of pre-synthesized SiO_2_ particles modified with -NH_3_^+^ groups with the complex [255,256,257]. It should be noted that most of these papers are aimed at obtaining materials specifically for biological applications (or at least mention them as possible), with only a few exceptions—Dechezelles et al. [246] and Nguyen et al. [258]. In the first paper, the authors obtained photonic (colloidal) crystals that exhibit luminescence depending on the angle of excitation radiation by arranging SiO_2_ particles doped with Cs_2_[{Mo_6_Br_8_}Br_6_] complex on a substrate [246]. In the second one, mesoporous silica particles functionalized with Cs_2_[{Mo_6_I_8_}(OCOC_2_F_5_)_6_] were used as model insulator material [258].

Aubert et al. [244] is a key paper in this field for two reasons: it was the first study on the biological properties of cluster-containing silica in vitro, and the exact binding modes realized between cluster and matrix were determined there. The materials themselves were prepared by the microemulsion method in the presence of Cs_2_[{Mo_6_Br_8_}Br_6_] complex (average particle size is ~45 nm) (Figure 17A). The obtained nanoparticles were studied by solid-state ^29^Si MAS NMR (Figure 17B). The data showed that during the material preparation, the complex is at least partially hydrolyzed, forming [{Mo_6_Br_8_}Br_6-x-y_(OH)_x_(H_2_O)_y_]^n^ units. As a result, two types of cluster-matrix interactions are realized—covalent bonds (Mo-O-Si) and hydrogen bonds between the apical ligands of the complex and the OH groups of silicon dioxide (Mo-OH_2_···O(H)-Si). Cytotoxicity studies showed that the particles have rather low dose-dependent toxicity towards the Caco-2 (colon adenocarcinoma cells) and MRC-5 (human lung fibroblasts) cell lines (Figure 17C,D). This article is the first in a series of studies demonstrating the biological applications of such materials.

For example, in [248] exploiting the same approach, the authors obtained solid silica particles with an average diameter of 50 nm containing Cs_2_[{Mo_6_I_8_}(OCOC_2_F_5_)_6_] clusters (Figure 18A,B). The particles were additionally modified with human transferrin protein (Tf) to enhance cellular uptake. Due to the long emission lifetimes (tens of microseconds), the obtained materials were approbated as agents for luminescence imaging using time-gated luminescence microscopy. Figure 18C shows a clear cluster-related emission in SKmel 28 cells (human melanoma cells) after ~10 µs delay.

In a series of our studies, an extended investigation of silica micro- and nanoparticles containing octahedral iodide molybdenum cluster complexes was carried out [245,247,250,251]. Microparticles (SMPs) with diameters of ~500 nm (Figure 19A) were obtained using the Stöber process, i.e., by ammonia hydrolysis of TEOS in acetone in the presence of [{Mo_6_I_8_}(NO_3_)_6_]^2−^, while nanoparticles (SNPs) with diameters of ~50 nm (Figure 19B) were obtained by the microemulsion method using the same reagents [245]. It was shown that despite similar emission properties, the materials photosensitize the singlet oxygen generation process with different efficiencies depending on their specific surface area, which is directly related to the particle size (Figure 19C).

Due to the low photosensitization efficiency, cluster-containing microparticles were used as luminescent carriers for protein delivery, allowing the track the position of the cargo without its oxidative destruction [247]. The SMP surface was modified with green fluorescent protein (GFP), which is unable to enter the cells in individual form. GFP-modified particles showed increased cytotoxicity against Hep-2 cells (human larynx carcinoma cells) compared to control samples, indicating successful modification and delivery of the protein into the cell. This was also confirmed by CLSM. One can see, that the red cluster emission completely overlaps with the green emission of GFP inside the cells (Figure 20A).

In contrast to SMPs, SNPs exhibited a high photosensitization efficiency, allowing them to be studied in photodynamic therapy. Indeed, incubation of the particles with Hep-2 cells in non-toxic concentrations followed by irradiation with white light (λ ≥ 400 nm) resulted in the appearance of apoptotic and dead cells (Figure 20B) [250]. It should be noted that the efficiency of the SNPs is comparable to that of the commercial PS Radachlorin^®^. The next step was to impart to this system a targeted delivery property [251]. For this purpose, the surface of SNPs was modified with the single domain antibody C7b specific for HER2/neu. To confirm the targeting property, cellular uptake was studied using CLSM on two cell lines—Hep-2 with low HER2 expression and SKBR3 (human breast cancer cells) overexpressing the receptor. In the case of SKBR3 cells, the red emission of the cluster appeared after only 15 min of incubation, indicating a higher rate of the SNPs uptake, thus confirming targeted delivery (Figure 20C). The in vivo PDT effect of the obtained system was evaluated in nude mice bearing grafted SKBR3 tumor. After intratumoral administration of the material, the tumor was irradiated with white light (λ ≥ 400 nm) for 30 min. After the third PDT treatment, the tumour’s signs of swelling and hyperemia were significantly reduced, while tissue necrosis increased (Figure 20D).

Another approach to obtain luminescent and PDT-active cluster-containing system was used in a series of studies conducted in a group of Dr. A.R. Mustafina [255,256,257]—binding of negatively charged clusters or cluster aggregates onto the positively charged amino-modified silica nanoparticles via electrostatic interactions. In [255], the authors obtained silica nanoparticles with an average diameter of ~60 nm decorated with ~3500 amino groups per NP using a microemulsion method (Figure 21A). To stabilize both the surface of the nanoparticles, which tend to aggregate in basic solution and the cluster complexes, which undergo rapid hydrolysis, the authors used the triblock copolymer L64 with the formula (PEO)_13_(PPO)_30_(PEO)_13_. Thus, SNPs decorated with K_2_[{Mo_6_I_8_}I_6_] (SN1) and (Bu_4_N)_2_[{Mo_6_I_8_}(CH_3_COO)_6_] (SN3) were prepared. Nanoparticles decorated with (Bu_4_N)_2_[{Mo_6_I_8_}(CH_3_COO)_6_] without the use of L64 (SN2) were also synthesized to investigate the importance of triblock copolymer utilization. TEM images revealed that after cluster sorption, significant outgrowths were observed on the silica surface (Figure 21B). According to the authors, ionic interactions between the clusters and the silica surface mainly occur in solution, leading to an increase in cluster concentration at the silica/water interface. Subsequent centrifugation switches the interaction mode from electrostatically driven cluster-silica interactions to self-assembly of cluster complexes at the silica/water interface. The resulting materials showed high stability in aqueous media, low cytotoxicity against MCF-7 cells, and a high rate of cellular uptake. Phototoxicity was investigated in cancerous MCF-7 and healthy HSF (human skin fibroblasts) cells. One can see that the materials exhibit high and selective photo-induced cytotoxicity against MCF-7 cells, while HSF cells remain almost unharmed even after irradiation (Figure 21C).

In [256], the authors introduced in the same system superparamagnetic iron oxide Fe_3_O_4_ particles (Figure 22A,B). The presence of the particles resulted in more stable ROS generation, which is most likely related to the higher stability in aqueous media and higher accumulation of the materials in the cells. According to the authors, this behavior can be explained by residual amounts of Fe^2+^ ions at the silica surface (Figure 22C).

## 7. Conclusions

As thoroughly discussed, the inorganic red/NIR emitters are strong rivals to organic ones in biomedical applications. Despite numerous promising luminophores of this type, in the review, we highlighted the most outstanding representatives, namely, lanthanide complexes and upconversion nanoparticles (UCNPs), quantum dots of different natures, ruthenium complexes, and transition halide metal cluster complexes. High photostability and, in certain cases, the presence of a set of interrelated properties (e.g., upconversion luminescence, photosensitization, radiopacity, photoactivated reactions, etc.) makes them useful as agents for diagnosis, therapy, or theranostics. However, low stability in aqueous media is considered to be one of the major drawbacks shared by all representatives of the entitled group, thus hindering their further development in biomedicine. Using simple and cheap methods, such as incorporation into the stable matrix protecting from the external environment, for stabilising unstable compounds is highly favorable. The most common matrix enormously widespread in the biomedical field is silicon dioxide, providing the materials not only stability and biocompatibility but also an additional opportunity to easily modify both the interior and exterior of the system, resulting in the combination of multiple properties that are rarely combined within an individual compound (e.g., address delivery, tracking via multiple techniques, and PDT effect). This allows the development of biocompatible multifunctional systems that can be tracked by multiple independent techniques, increasing detection accuracy and/or have targeted combined actions induced by external stimuli, increasing therapeutic efficacy. Although such systems are still only concepts in scientific papers, their potential is too high to be ignored, which must lead to their application in practical biology and medicine in the future.

## Figures and Tables

**Figure 1 materials-16-05869-f001:**
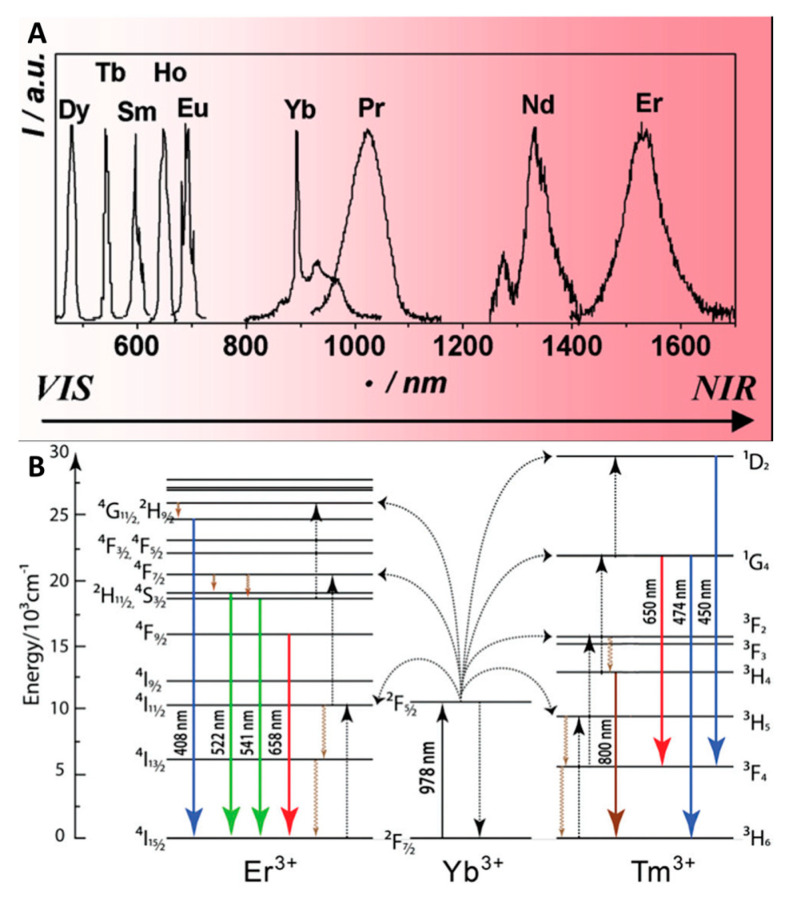
Emission spectra of a series of lanthanide complexes (**A**). Adapted from [63] with permission from the American Chemical Society. The energy transfer processes of upconversion nanoparticles co-doped with Yb^3+^, Er^3+^/Tm^3+^ under NIR laser irradiation (**B**). Adapted from [64] with permission from the Royal Society of Chemistry.

**Figure 2 materials-16-05869-f002:**
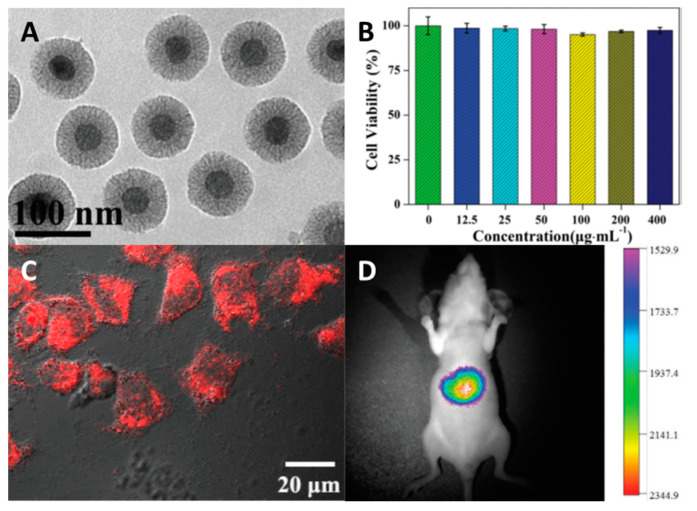
TEM image of UCNPs@mSiO_2_-Eu(dbm)_4_ (**A**); In vitro cell viabilities of HeLa cells incubated with UCNPs@mSiO_2_-Eu(dbm)_4_ (**B**); Confocal imaging of HeLa cells incubated with UCNPs@mSiO_2_-Eu(dbm)_4_ (**C**); In vivo upconversion luminescence images of a nude mouse acquired at 1 h after intravenous injection of UCNPs@mSiO_2_-Eu(dbm)_4_ (**D**). Adapted from [79] with permission from the Royal Society of Chemistry.

**Figure 3 materials-16-05869-f003:**
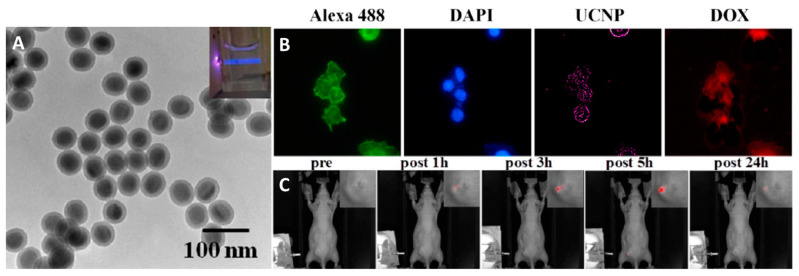
TEM image of silica-coated NaYF_4_:Yb,Tm nanoparticles (**A**); Confocal imaging of HeLa cells incubated with caged folate-PEGylated UCNPs@SiO_2_-DOX by exposure of a 980 nm diode laser (**B**); Time course upconversion NIR luminescence (λ_em_ = 800 nm) images of caged folate-PEGylated UCNPs@SiO_2_-DOX nanoparticles after NIR laser irradiation (**C**). Adapted from [76] with permission from the American Chemical Society.

**Figure 4 materials-16-05869-f004:**
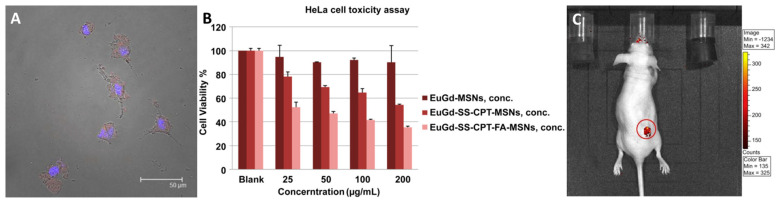
Confocal imaging of HeLa cells incubated with FA-modified europium-containing mSiO_2_ (**A**); Cytotoxicity of europium-containing mSiO_2_ studied on HeLa cells (**B**)*;* In vivo PL imaging of mice after subcutaneous injection of FA-modified europium-containing mSiO_2_ (λ_ex_ = 430 nm) (**C**). Adapted from [126] with permission from the American Chemical Society.

**Figure 5 materials-16-05869-f005:**
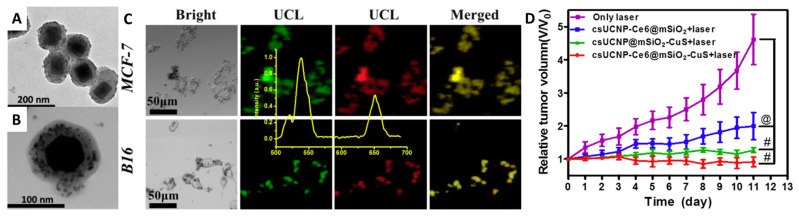
TEM images of UCNPs-Ce_6_@mSiO_2_-CuS (**A**, **B**); Confocal imaging of MCF-7 cells (up) and B16 cells (bottom) incubated with the UCNPs-Ce_6_@mSiO_2_-CuS (**C**); Tumor growth curves of B16 tumor-bearing mice after various treatments (**D**). Adapted from [94] with permission from Elsevier.

**Figure 6 materials-16-05869-f006:**
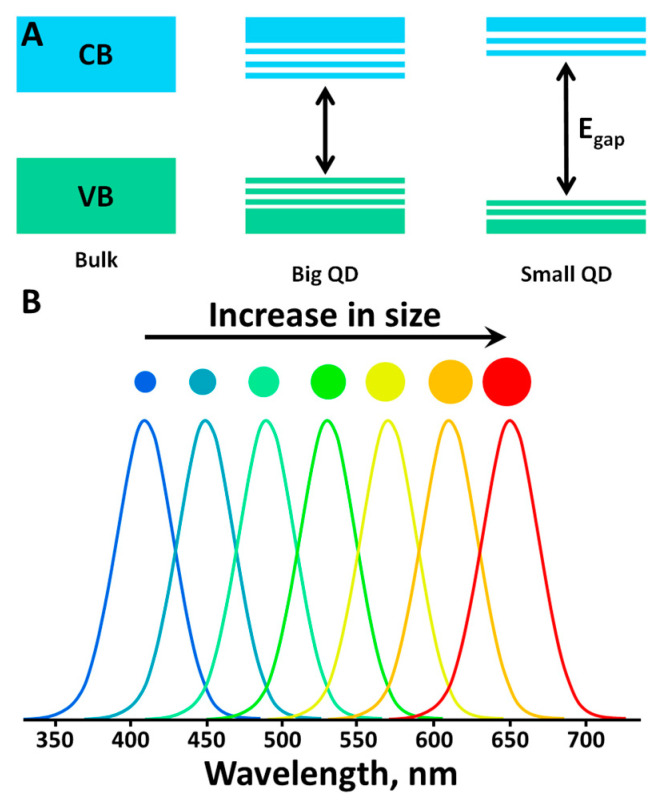
Change in bandgap upon reduction of the physical size of the semiconductor (**A**). Emission spectra of QDs as a function of size (**B**).

**Figure 7 materials-16-05869-f007:**
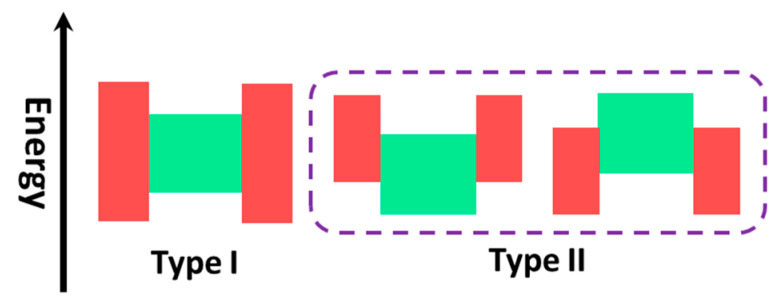
Core-shell QD types (*core—green, shell—red*).

**Figure 8 materials-16-05869-f008:**
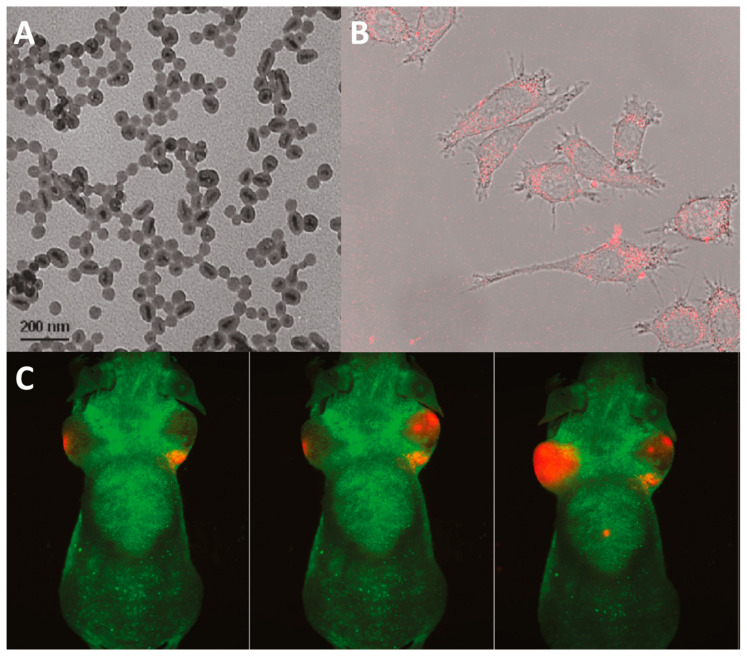
TEM image of SiO_2_-coated QRs with shell thickness of ~10 nm (**A**); Confocal Images of Panc 1 cell line treated with an aqueous dispersion of SiO_2_-coated QRs with a shell thickness of ~10 nm (**B**); Whole-body imaging of tumor-bearing mice intratumoraly injected with SiO_2_-coated QRs with shell thickness of ~10 nm at 1 (left), 5 (middle), and 10 (right) min postinjection (**C**). Adapted from [152] with permission from the American Chemical Society.

**Figure 9 materials-16-05869-f009:**
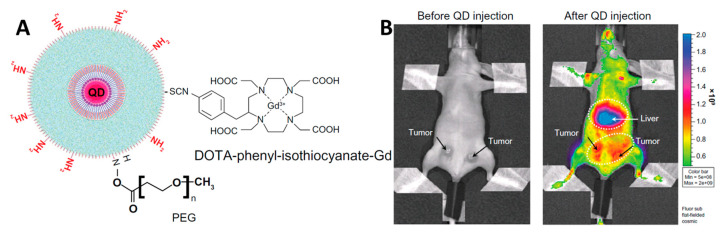
Model structure of PEGylated multimodal silica-shelled quantum dots (**A**). Adapted from [156] with permission from the American Chemical Society. Fluorescent imaging of tumor by angiogenesis in the anesthetized mouse injected intravenously with PEG1100-grafted multimodal QD (λ_ex_ = 450 ± 30 nm, λ_em_ = 650 nm) (**B**). Adapted from [157] with permission from DovePress.

**Figure 10 materials-16-05869-f010:**
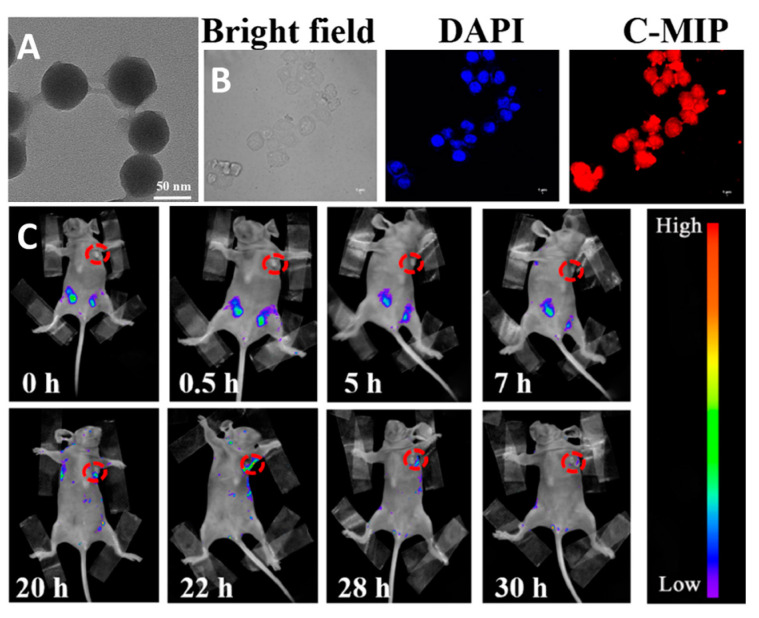
TEM image of C-MIP (**A**); Confocal fluorescence imaging of HeLa cells incubated with C-MIP (**B**); In vivo fluorescence imaging and distribution of C-MIP with time in nude mice bearing HeLa tumors after intraperitoneal injection (Red circles are the tumor sites) (**C**). Adapted from [155] with permission from Springer.

**Figure 11 materials-16-05869-f011:**
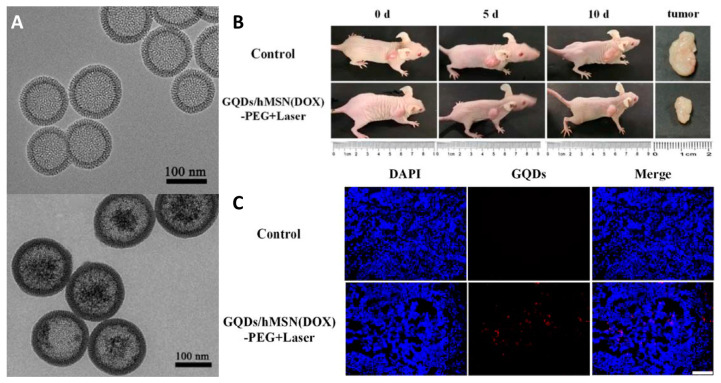
TEM image of mSiO_2_ and GQDs/mSiO_2_ (**A**); In vivo anticancer therapy of the material (635 nm, 300 mW cm^−2^, 10 min) (**B**); The distribution of GQDs/mSiO_2_ in the tumor by observing tumor frozen section using a fluorescence microscope (**C**). Adapted from [154] with permission from Elsevier.

**Figure 12 materials-16-05869-f012:**
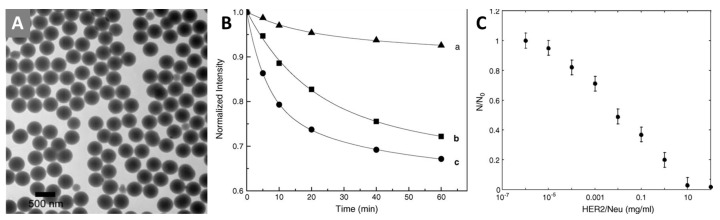
TEM image of Ru(phen)_3_ containing silica particles (d ~ 400 nm) (**A**); Photostability measurements: Ru(phen)_3_^3+^ containing silica particles (▲), silica particles prepared by loading Ru(phen)_3_ into pre-prepared silica particles (■), and a solution of 0.1 mM Ru(phen)_3_^3+^ (●) (**B**); Dependence of the number of fluorescent particles immobilized to anti-HER2/neu-modified glass slides on the concentration of free HER2/neu in the analyte solution (**C**). Adapted from [178] with permission from Elsevier.

**Figure 13 materials-16-05869-f013:**
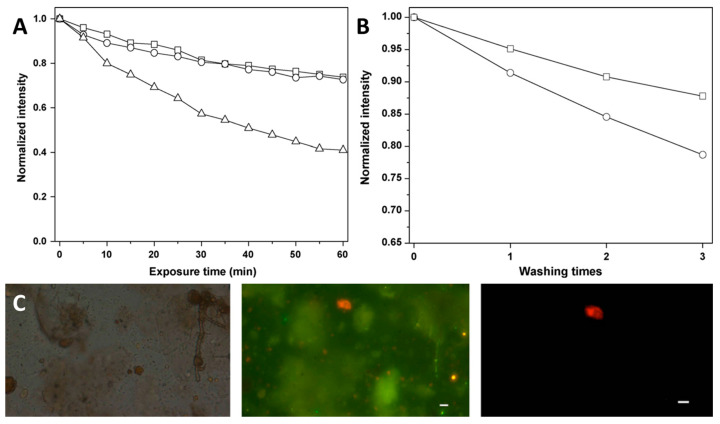
Photostability measurements: free [Ru(NH_2_-phen)_3_]^2+^ complex (△), Ru(II) complex covalently bound nanoparticles (□), and physically encapsulated nanoparticles (○) (**A**); Leakage experiments: the Ru(II) complex covalently bound nanoparticles (□) and physically encapsulated nanoparticles (○) (**B**); Bright-field (left), luminescence (middle) and time-gated luminescence (right) images of *Giardia lamblia* stained by the nanoparticle-labeled SA in a complex environmental water sample (Scale bars, 10 µm) (**C**). Adapted from [201] with permission from Elsevier.

**Figure 14 materials-16-05869-f014:**
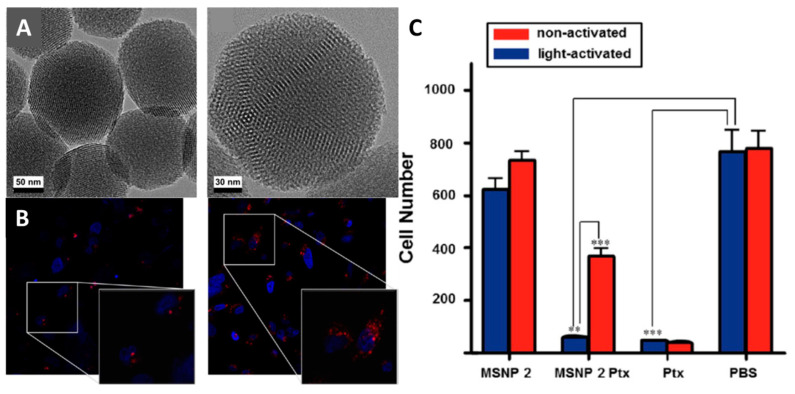
TEM image of Ruthenium-containing mSiO_2_ NPs (**A**); Merged confocal images of MDA-MB-468 breast cancer cells treated with non-activated (left) or light-activated (right) ruthenium-containing mSiO_2_ NPs (**B**); Cell survival of MDA-MB-231 cells with or without light activation. The data are presented as the mean and the standard error of the mean (SEM) of three experiments ((***) *p* < 0.001, (**) *p* < 0.01) (**C**). Adapted from [207] with permission from the American Chemical Society.

**Figure 15 materials-16-05869-f015:**
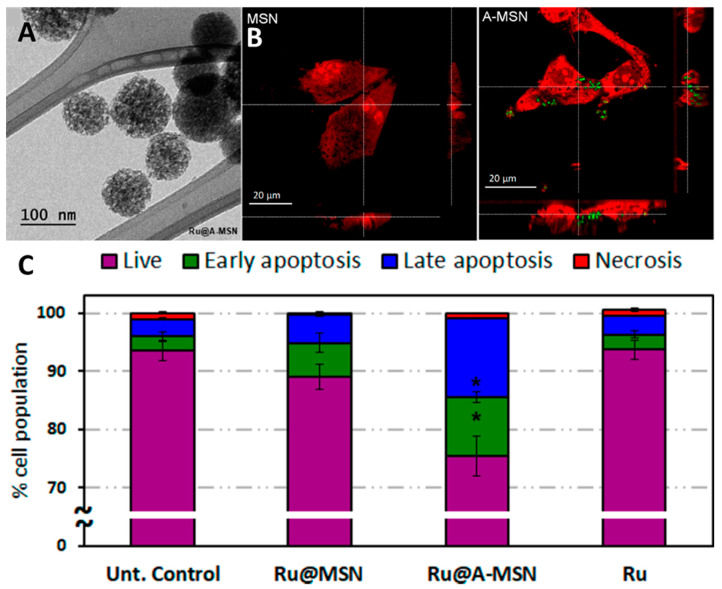
TEM image of Ru-containing amino-modified mSiO_2_ NPs (**A**); Confocal images of U87 human glioblastoma cells treated with unmodified (left) or amino-modified (right) Ru-containing mSiO_2_ NPs (**B**); Dual Annexin V-FITC/PI flow cytometric analysis of U87 cells treated with Ru-containing amino-modified mSiO_2_ NPs. Statistical significance was calculated using an unpaired t test (* *p* < 0.05) from two independent experiments (n = 2 replicates) (**C**). Adapted from [195] with permission from the American Chemical Society.

**Figure 16 materials-16-05869-f016:**
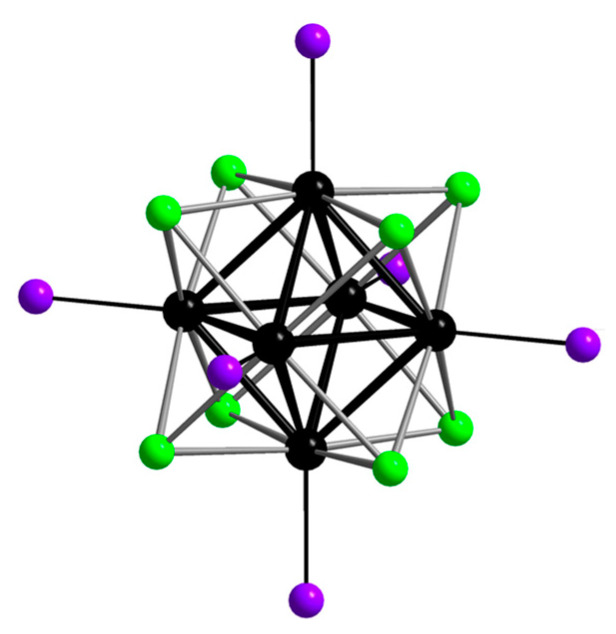
Representative structure of octahedral metal cluster complexes [{M_6_X_8_}L_6_]^n^. Color code: black—metal (Mo or W), green—inner ligands X, purple—apical ligands L.

**Figure 17 materials-16-05869-f017:**
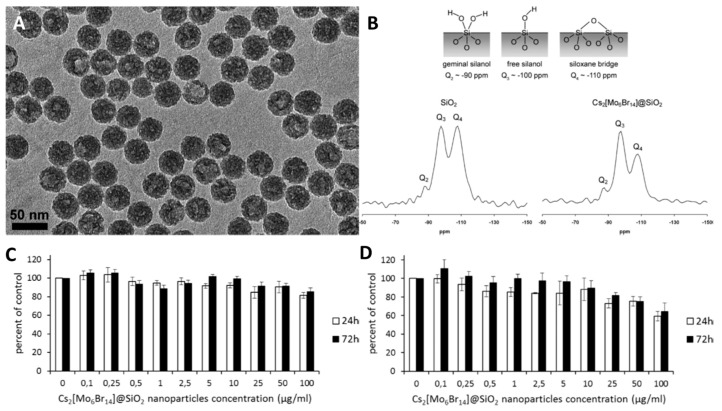
TEM image of Cs_2_[{Mo_6_Br_8_}Br_8_]@SiO_2_ nanoparticles (**A**); ^29^Si MAS NMR spectra of SiO_2_ (left) and Cs_2_[{Mo_6_Br_8_}Br_8_]@SiO_2_ (right) nanoparticles (**B**); Viability of Caco-2 (**C**) and MRC-5 (**D**) cells treated with Cs_2_[{Mo_6_Br_8_}Br_8_]@SiO_2_ nanoparticles. Adapted from [244] with permission from the American Chemical Society.

**Figure 18 materials-16-05869-f018:**
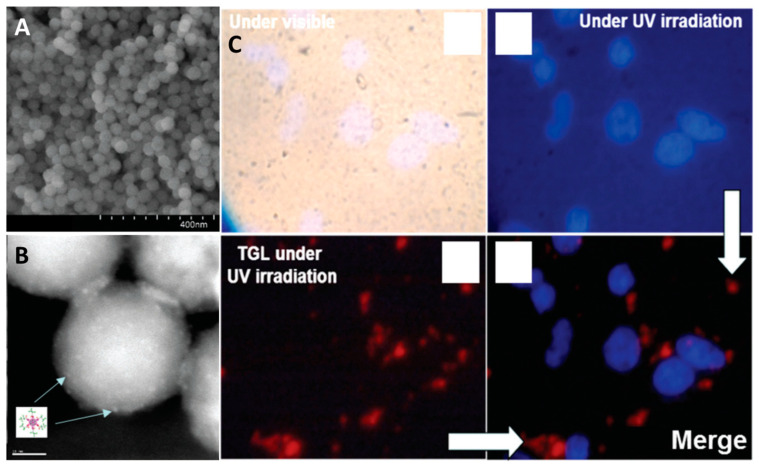
SEM (**A**) and HAADF-STEM (**B**) images of Cs_2_[{Mo6I_8_}(OCOC_2_F_5_)_6_]@SiO_2_; Images of the Tf-nanoparticles internalized by SKmel 28 cancer cells using different modes of the time-gated luminescence microscope (**C**). Adapted from [248] with permission from the Royal Society of Chemistry.

**Figure 19 materials-16-05869-f019:**
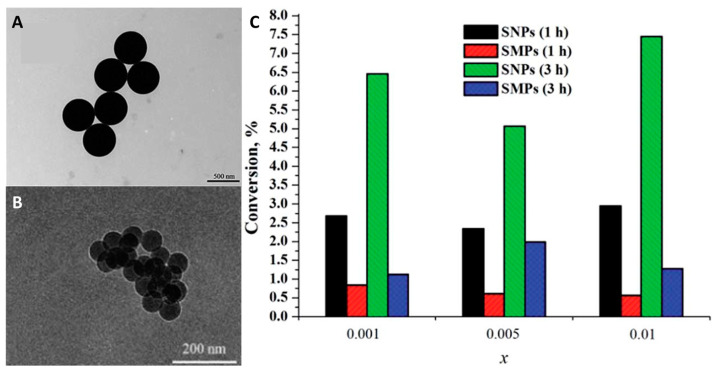
TEM images of the cluster containing micro—(**A**) and nanoparticles (**B**); Singlet oxygen generation by cluster containing micro- and nanoparticles (**C**). Adapted from [245] with permission from the Royal Society of Chemistry.

**Figure 20 materials-16-05869-f020:**
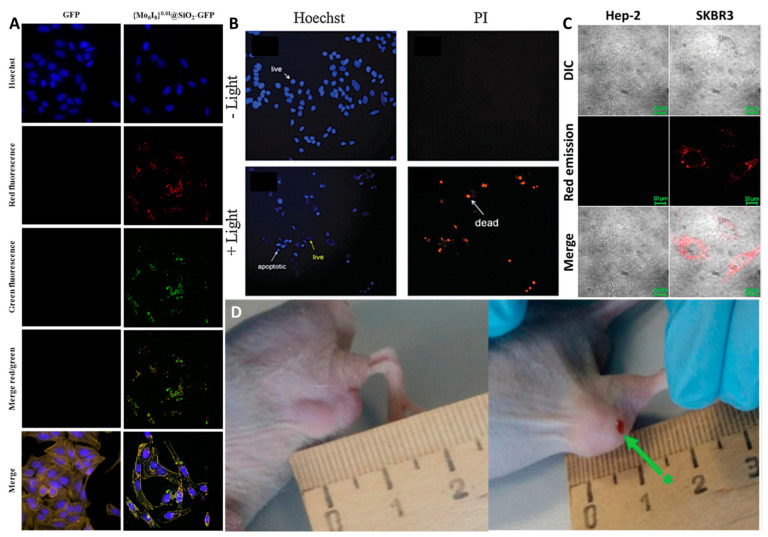
Cellular uptake of free GFP and GFP-modified SMPs by fluorescent microscopy (**A**). Adapted from [247] with permission from Elsevier. Effect of SNPs on Hep-2 cells before (top) and after (bottom) photoirradiation determined by dual staining with Hoechst 33342/PI (**B**). Adapted from [250] with permission from the Royal Society of Chemistry. Confocal microscopic images of Hep-2 and SKBR3 cells incubated with C7b-modified SNPs for 15 min (**C**); In vivo PDT effect (λ ≥ 400 nm) of C7b-modified SNPs in a nude mouse bearing grafted SKBR3 tumor (**D**). Adapted from [251] with permission from the Royal Society of Chemistry.

**Figure 21 materials-16-05869-f021:**
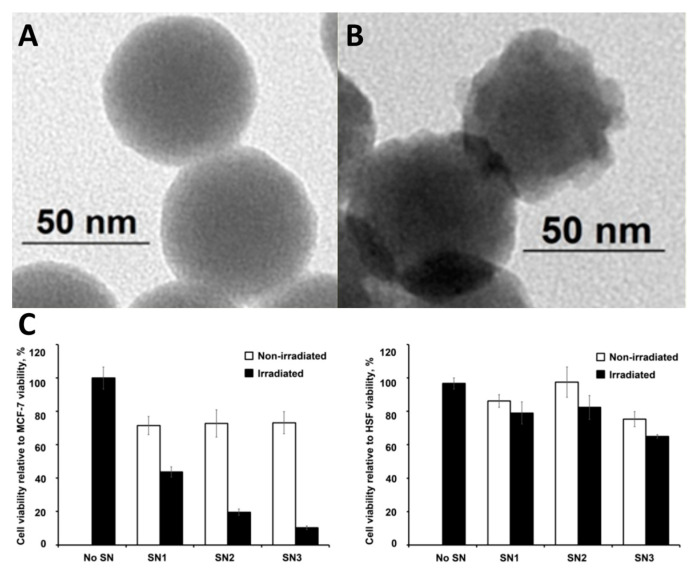
TEM images of the pure SNPs (**A**) and cluster-containing SNPs (**B**); The cell viability of irradiated and non-irradiated MCF-7 (left) and HSF (right) cell lines treated with different types of cluster-containing SNPs (**C**). Adapted from [255] with permission from Elsevier.

**Figure 22 materials-16-05869-f022:**
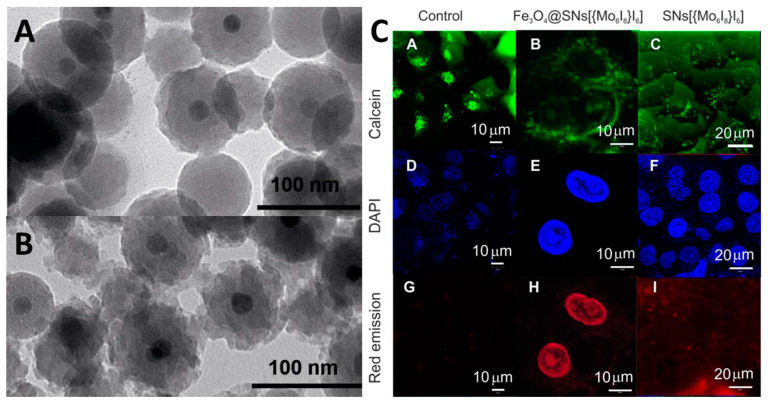
TEM images of the pure Fe_2_O_3_@SNPs (**A**) and cluster-containing Fe_2_O_3_@SNPs (**B**); CLSM images of M-Hela cells incubated with Fe_2_O_3_@SNPs and cluster-containing Fe_2_O_3_@SNPs (**C**). Adapted from [256] with permission from Elsevier.

## Data Availability

Data sharing not applicable.

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
