# Peer review of "Silica-Based Materials Containing Inorganic Red/NIR Emitters and Their Application in Biomedicine"

_materials, 2023, doi:10.3390/ma16175869_

Round 1

Reviewer 1 Report

Comments to Authors

1.       Abstract; due to its properties? Mentioned it.

2.       Abstract; classified the methods. Which methods?

3.       Highlights some key points.

4.       Write keywords in alphabetical order. Remove UCNP and PDT.

5.       Please mention the Graphical abstract.

6.       Introduction, start it from 1. Revise it and could be elaborated in terms of what other mechanism has been used in previous or other related studies?

7.       Add the impact of current work on industry and future research.

8.       Ln-containing silica-based materials, QD-containing silica-based materials. Where is the sub-heading?

9.       5. Ruthenium complexes. Revise it and make it short and clear.

10.    For °C, Figure, Table, % mentioned same format throughout the manuscript.

11.    Authors need to improve the problem statement in the introduction section. Cite the latest literature data.

12.    Why and how the said parameters were selected for this work? More specific details needed to be added with the use of the latest reference. Particularly section 6. Octahedral metal cluster complexes.

13.    In your discussion section, please link your empirical results with a broader and deeper literature review.

14.    Figure 13. Mentioned the original dimension.

15.    Do not cite more than three references at the same time [14-17, 59-77, 70-77, etc].

16.    Use Endnote or Mendeley for reference. The current format is not according to the prestigious journal.

17.    Explain the conclusion in more detail.

Authors need to increase the literature and problem statement from the current recent papers such as.

v  https://doi.org/10.1007/s10924-022-02561-8.

v   doi: 10.3390/bioengineering9080406

The author should want to check the whole manuscript with the birds' eyes.

Author Response

Reviewer 1

Thank you for your positive feedback and relevant suggestions, which we address below.

  1. Abstract; due to its properties? Mentioned it.

We have added a list of properties.

  1. Abstract; classified the methods. Which methods?

We have rephrased this sentence.

  1. Highlights some key points.

We have carefully revised the manuscript and highlighted some major points.

  1. Write keywords in alphabetical order. Remove UCNP and PDT.

It is done.

  1. Please mention the Graphical abstract.

Done.

  1. Introduction, start it from 1. Revise it and could be elaborated in terms of what other mechanism has been used in previous or other related studies?

Introduction section was partially rewritten and supported with additional recent references.

  1. Add the impact of current work on industry and future research.

Added.

  1. Ln-containing silica-based materials, QD-containing silica-based materials. Where is the sub-heading?

Sub-headings for these chapters without numeration were used according to journal style.

  1. 5. Ruthenium complexes. Revise it and make it short and clear.

Thank you for your suggestion. This section was revised and rewritten in order to make in more clear.

  1. For °C, Figure, Table, % mentioned same format throughout the manuscript.

Done.

  1. Authors need to improve the problem statement in the introduction section. Cite the latest literature data.

The problem was supported with additional recent references.

  1. Why and how the said parameters were selected for this work? More specific details needed to be added with the use of the latest reference. Particularly section 6. Octahedral metal cluster complexes.

Thank you for your comment. The red/NIR emitters were highlighted in this review due to their high potential in biomedical field. As it mentioned in Introduction, the emitter chosen in this work have some drawbacks mainly related to solubility and stability. Thus, they need stabilization. Despite SiO2 is one of the most widespread type of matrices, there are no reviews combining stabilization and biomedical application of silica-based materials containing inorganic red/NIR emitters. Indeed, there are some other luminophores, which do not require stabilization, but they are out of focus of this paper. The references used in the review are mainly published in last 10-15 years. In particular, in section 6 Octahedral metal cluster complexes we used references mainly published in 2016-2023. In the section Cluster-containing silica-based materials all existing papers on this topic were used.

  1. In your discussion section, please link your empirical results with a broader and deeper literature review.

We have carefully revised the manuscript and supported our suggestions with additional relevant references.

  1. Figure 13. Mentioned the original dimension.

Done.

  1. Do not cite more than three references at the same time [14-17, 59-77, 70-77, etc].

Thank you for you comment. We thoroughly optimized references order to make them shorter.

  1. Use Endnote or Mendeley for reference. The current format is not according to the prestigious journal.

Endnote 20 software was used for references.

  1. Explain the conclusion in more detail.

The conclusion was modified.

Authors need to increase the literature and problem statement from the current recent papers such as.

https://doi.org/10.1007/s10924-022-02561-8, doi: 10.3390/bioengineering9080406

Thank you for your comment. We have carefully read the suggested papers and used them as inspiration to increase the quality our review.

Reviewer 2 Report

Overall, the manuscript presents valuable insights into the realm of red/NIR emitters and their incorporation into silica-based materials for biomedical applications. The authors have effectively highlighted the significance of this field. However, in order to enhance the quality of the manuscript, I recommend addressing few comments:

1.     The resolution of all figures should be increased to ensure clarity and facilitate better understanding.

2.     To further enrich the discussion, it would be beneficial to include a brief section on metal oxide nanoparticles exhibiting NIR emission, their advantages, and disadvantages. Specifically, take reference of the work on MgO nanoparticles emitting in the NIR region and their utilization in bioimaging [Nanomaterials 2021, 11(3), 695 and Journal of Alloys and Compounds, 2021, 876, 160175] could provide a more comprehensive perspective on the topic.

3.     Additionally, there is room for improvement in the English language expression throughout the manuscript.

Extensive modification is needed.

Author Response

Reviewer 2

Overall, the manuscript presents valuable insights into the realm of red/NIR emitters and their incorporation into silica-based materials for biomedical applications. The authors have effectively highlighted the significance of this field. However, in order to enhance the quality of the manuscript, I recommend addressing few comments:

Thank you for your positive feedback and relevant suggestions, which we address below.

  1. The resolution of all figures should be increased to ensure clarity and facilitate better understanding.

The resolution of the figures taken from the original papers are limited by initial quality, thus we can not increase it. Nevertheless, we believe, that overall quality of the figures in original docx file is sufficient, and reviewer opinion can be caused by compression of the file during converting in pdf.

  1. To further enrich the discussion, it would be beneficial to include a brief section on metal oxide nanoparticles exhibiting NIR emission, their advantages, and disadvantages. Specifically, take reference of the work on MgO nanoparticles emitting in the NIR region and their utilization in bioimaging [Nanomaterials 2021, 11(3), 695 and Journal of Alloys and Compounds, 2021, 876, 160175] could provide a more comprehensive perspective on the topic.

Thank you for your valuable comment! We have made a literature search on this topic and unfortunately did no find much papers studying silica-based materials containing such oxide systems. Nevertheless, we added these references along with those related to gold nanoclusters in Introduction section as a representation of other types of inorganic red/NIR emitters promising for this field.

  1. Additionally, there is room for improvement in the English language expression throughout the manuscript.

We have carefully revised the manuscript in order to improve the quality of English used.

Reviewer 3 Report

Comments:

In the manuscript titled “Silica-based materials containing inorganic red/NIR emitters and their application in biomedicine” the authors have discussed inorganic red/NIR emitters, in ruthenium complexes, quantum dots, lanthanide compounds, and octahedral cluster complexes of molybdenum and tungsten, exhibit excellent emission in the desired region, with additional functional properties, such as photosensitization of the singlet generation process, up-conversion luminescence, photoactivated effects, and so on. The surface modification provides room for creativity in the development of various multifunctional diagnostic/therapeutic platforms which overcome the limitation of instability in aqueous media. The authors collected and classified the methods of obtaining and application areas of the silica-based materials containing the above compounds. The details and discussion of the literature are quite vast and encouraging; however, I think the manuscript has the following significant issues.

Minor issues

1.     Line 33-34, some of the organisms grow under red light conditions. So, it is not correct to say a very little impact.

2.     Lines 47-51, could be rewritten in a better way, and seem confusing while reading the first time.

3.     Line 62, organic dyes usually don't show narrow emission bands, mostly broad.

4.     Lack of enough references at multiple places. For e.g. lines 66, 70, 77, 572, etc.

5.     Line 387, ‘rode’ instead of rod.

6.     Line 564, rather than writing ‘chapters’, would prefer ‘sections’.

7.     Line 568, ‘In a large number of works’ sounds non-scientific. The authors could write as ‘In a series of studies’.

8.     Line 578, no space between above mentioned.

9.     Figure 13 caption, caption shows filled symbols whereas figures show no fill.

10.  Some of the figure captions do not describe well all of the figures, for e.g., figure 13 (C).

11.  Figure 16 caption, no details about colors showing different atoms.

12.  There is a problem with the tense used at different places.

Major issues

1.     Line 227 and 231, initially the discussion is about the Ln complex showing emission 800-1700 nm region, later the CLSM data shows emission in the 600-700 nm region. Was the material different or it shows up conversion luminesce? If so, the discussion should be clear to understand.

2.     At multiple places, the research group is used as an article. For e.g. line 291, ‘In Sun et al.’, line 386, ‘is Kumar et al.’, line 418, ‘In Bakalova et al.’, line 450, ‘In Zhang et al.’etc. The authors should represent them as a group. At multiple places, the choice of words seems inappropriate. For e.g., ‘object’ is used in line 771 rather than a material or compound, ‘work’ is used rather than ‘studies’, etc.

3.     Line 398, and figure 8, the TEM image looks like it has a size distribution and most of them looks spherical rather than rods as discussed in the text. Does that true? Could the authors explain how did they segregate them?

4.     Line 433, does the coating with PEG1100 go away in aqueous and different pH environment?

5.     Line 440, what is the reason bind liver showing maximum intensity? Is there any kind of interaction of coated QDs with the local environment or its binding specificity?

6.     Line 469, from the images it looks like the emission stays only for a few hours. Does that mean the material degraded quickly inside the cancerous tissue?

7.     Line 482, this is more general question. Two photon-excitation needs really high power and fluence of light source. How did the authors make sure that there is no photo damage of cells?

8.     Line 629, do the authors want to say it is purchased directly?

9.     Line 648, the ligand exchange is crucial for drug delivery, where nitrile group coordinates with Ru after releasing water molecule. However, line 658 says, formation of water complex which further used for drug delivery. Do both sentences talk about different compounds?

10.  Line 657, the water complex shows a decrease in fluorescence intensity. Is it due to aggregation?

11.  Line 746 can be rewritten for more clarification. Does that mean different light sources can be used to excite these compounds?

Overall, there are a lot of studies cited in the paper, lacking clarity in the discussion of results and experiments to support them. In the discussion, several concepts are anticipated without the presentation of the data or citing references, and therefore it is very confusing. Addressing these points would enhance the clarity and quality of the manuscript.

English looks confusing and wobbly in multiple places. Sometimes the choice of words doesn’t seem appropriate.

Author Response

Reviewer 3

In the manuscript titled “Silica-based materials containing inorganic red/NIR emitters and their application in biomedicine” the authors have discussed inorganic red/NIR emitters, in ruthenium complexes, quantum dots, lanthanide compounds, and octahedral cluster complexes of molybdenum and tungsten, exhibit excellent emission in the desired region, with additional functional properties, such as photosensitization of the singlet generation process, up-conversion luminescence, photoactivated effects, and so on. The surface modification provides room for creativity in the development of various multifunctional diagnostic/therapeutic platforms which overcome the limitation of instability in aqueous media. The authors collected and classified the methods of obtaining and application areas of the silica-based materials containing the above compounds. The details and discussion of the literature are quite vast and encouraging; however, I think the manuscript has the following significant issues.

Thank you for your positive feedback and relevant suggestions, which we address below.

Minor issues

  1. Line 33-34, some of the organisms grow under red light conditions. So, it is not correct to say a very little impact.

Corrected.

  1. Lines 47-51, could be rewritten in a better way, and seem confusing while reading the first time.

The sentences were rewritten.

  1. Line 62, organic dyes usually don't show narrow emission bands, mostly broad.

Corrected.

  1. Lack of enough references at multiple places. For e.g. lines 66, 70, 77, 572, etc.

We have added references to support our suggestions.

  1. Line 387, ‘rode’ instead of rod.

Corrected.

  1. Line 564, rather than writing ‘chapters’, would prefer ‘sections’.

corrected

  1. Line 568, ‘In a large number of works’ sounds non-scientific. The authors could write as ‘In a series of studies’.

Corrected.

  1. Line 578, no space between above mentioned.

Corrected.

  1. Figure 13 caption, caption shows filled symbols whereas figures show no fill.

Corrected.

  1. Some of the figure captions do not describe well all of the figures, for e.g., figure 13 (C).

Corrected.

  1. Figure 16 caption, no details about colors showing different atoms.

Details were added in the Figure caption.

  1. There is a problem with the tense used at different places.

We have carefully revised the manuscript in order to improve the quality of English used.

Major issues

  1. Line 227 and 231, initially the discussion is about the Ln complex showing emission 800-1700 nm region, later the CLSM data shows emission in the 600-700 nm region. Was the material different or it shows up conversion luminesce? If so, the discussion should be clear to understand.

Thank you for your comment. Indeed, emission range was indicated incorrectly. Also, the system studied in terms of biological investigations was specified (UCNPs@mSiO2-Eu(dbm)4) for better understanding.

  1. At multiple places, the research group is used as an article. For e.g. line 291, ‘In Sun et al.’, line 386, ‘is Kumar et al.’, line 418, ‘In Bakalova et al.’, line 450, ‘In Zhang et al.’etc. The authors should represent them as a group. At multiple places, the choice of words seems inappropriate. For e.g., ‘object’ is used in line 771 rather than a material or compound, ‘work’ is used rather than ‘studies’, etc.

The text was corrected according to reviewer’s comment.

  1. Line 398, and figure 8, the TEM image looks like it has a size distribution and most of them looks spherical rather than rods as discussed in the text. Does that true? Could the authors explain how did they segregate them?

According to the authors of the paper discussed (doi:10.1021/cm902610f) all of the particles have rod-like shape. Indeed, there are some spherical particles in TEM images. Nevertheless, according to TEM images of initial quantum dots there are no spherical particles. Most likely, these SiO2-coated QRs are looks like spherical because of spatial orientation (pointed at us).

  1. Line 433, does the coating with PEG1100 go away in aqueous and different pH environment?

PEG1100 is covalently bounded to amino groups at QD surface, so in tested conditions it should be stable enough.

  1. Line 440, what is the reason bind liver showing maximum intensity? Is there any kind of interaction of coated QDs with the local environment or its binding specificity?

According to the authors and other investigations, polymer-coated quantum dots in general tend to accumulate predominantly in the liver. The main role of liver and kidneys is filtering and excretion, thus these particles are digested through liver.

  1. Line 469, from the images it looks like the emission stays only for a few hours. Does that mean the material degraded quickly inside the cancerous tissue?

The system described in this work is modified to be selective for epidermal growth factor receptor (EGFR). The emission arises from carbon quantum dots, which possess good photostability. Thus, after administration we observe bright emission from kidneys indicating rapid excretion of the main part of the nanoparticles. Further observations indicate gradual accumulation of the particles remaining in blood stream in tumor spot caused by selectivity property.

  1. Line 482, this is more general question. Two photon-excitation needs really high power and fluence of light source. How did the authors make sure that there is no photo damage of cells?

Depending on the efficiency of two-photon absorption, low power emission can be enough to excite the substance. The power of laser irradiation (635 nm) used was 20 mW/cm2 for in vitro investigations and 300 mW/cm2 for tumor treatment in mice. The safety of these conditions has been confirmed by the authors (10.1016/j.jddst.2020.102127). The conditions used were indicated in Figure caption for clarity.

  1. Line 629, do the authors want to say it is purchased directly?

Prior to covalent bonds formation phenanthroline ligand of initial complex [Ru(NH2-phen)3](PF6)2 (NH2-phen is 5-amino-1,10-phenanthroline) was modified with silanol group.

  1. Line 648, the ligand exchange is crucial for drug delivery, where nitrile group coordinates with Ru after releasing water molecule. However, line 658 says, formation of water complex which further used for drug delivery. Do both sentences talk about different compounds?

Under the synthetic conditions chosen by the authors, the coordination of the initial complex to the MCM-41 particles surface occurs by the substitution of water ligand for nitrile group (silica was preliminary modified with benzonitrile groups). If porous silica is preliminary soaked with Paclitaxel® cytostatic drug, it can be blocked within the pores by the coordinated ruthenium complex. Further irradiation of the sample cause photoinduced release of the complex followed by the release of Paclitaxel®. Low hydrolytic stability of the compound results in gradual hydrolysis of the complex released with the formation of non-luminescent aqua-complexes. Nevertheless, within the cells the complex released interacts with DNA molecule, which results in the increase in luminescence, as it shown in Figure 14B.

  1. Line 657, the water complex shows a decrease in fluorescence intensity. Is it due to aggregation?

A decrease in fluorescence intensity is caused by the hydrolysis of the released complex resulting in the formation of non-luminescent aqua complex.

  1. Line 746 can be rewritten for more clarification. Does that mean different light sources can be used to excite these compounds?

The sentence was rewritten.

Overall, there are a lot of studies cited in the paper, lacking clarity in the discussion of results and experiments to support them. In the discussion, several concepts are anticipated without the presentation of the data or citing references, and therefore it is very confusing. Addressing these points would enhance the clarity and quality of the manuscript.

Round 2

Reviewer 3 Report

In the manuscript titled “Silica-based materials containing inorganic red/NIR emitters and their application in biomedicine,” the efforts by the authors in revising the manuscript are appreciated.

Overall, the authors made necessary revisions to the manuscript and the manuscript is OK for publication. 

Looks fine, but there is a window for improvement.